# GPG: A Simple and Strong Reinforcement Learning Baseline for Model Reasoning

**Xiangxiang Chu, Hailang Huang, Xiao Zhang, Fei Wei, Yong Wang**
AMAP, Alibaba Group
⌂ Project Page: https://github.com/AMAP-ML/GPG

## Abstract

Reinforcement Learning (RL) can directly enhance the reasoning capabilities of large language models without extensive reliance on Supervised Fine-Tuning (SFT). In this work, we revisit the traditional Policy Gradient (PG) mechanism and propose a minimalist RL approach termed Group Policy Gradient (GPG). Unlike conventional methods, GPG directly optimizes the original RL objective, thus obviating the need for surrogate loss functions. By eliminating the critic and reference models, avoiding KL divergence constraints, and addressing the advantage and gradient estimation bias, our approach significantly simplifies the training process compared to Group Relative Policy Optimization (GRPO). Our approach achieves superior performance without relying on auxiliary techniques or adjustments. As illustrated in Figure 1, extensive experiments demonstrate that our method not only reduces computational costs but also consistently outperforms GRPO across various unimodal and multimodal tasks.

## 1 Introduction

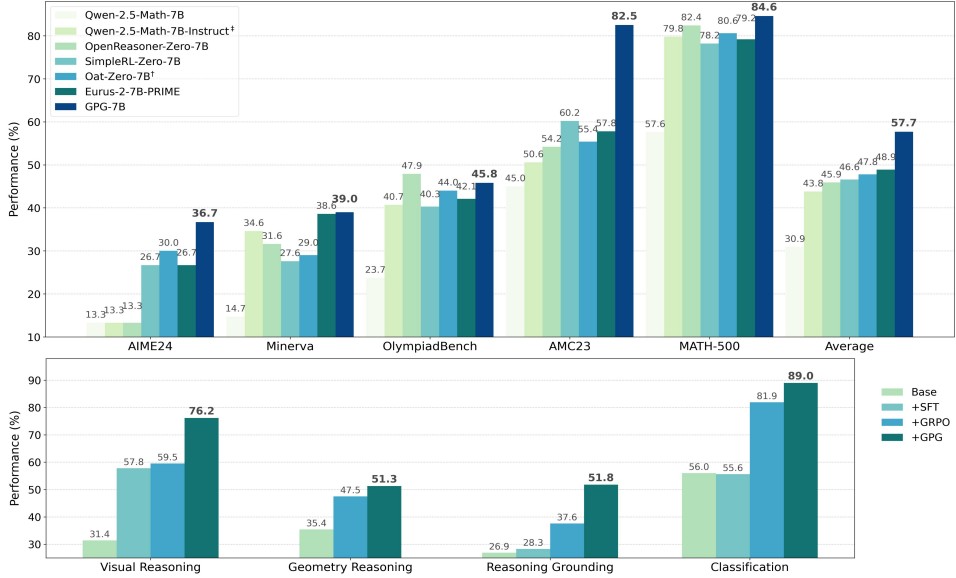

Figure 1: Performance comparison on unimodal reasoning tasks, with extended validation on multimodal reasoning. (**Top**) GPG achieves substantial performance gains over state-of-the-art (SOTA) baselines across diverse mathematical benchmarks, demonstrating its core effectiveness for linguistic reasoning. (**Bottom**) The method also generalizes robustly to multi-modal settings, outperforming other RL methods and further validating its broad applicability.

Large Language Models (LLMs) have achieved substantial advancements, progressively narrowing the gap towards achieving Artificial General Intelligence (AGI) (OpenAI, 2024; Guo et al., 2025; Bai et al., 2025; Wu et al., 2024; Yao et al., 2024). Recently, LLMs, exemplified by OpenAI o1 (OpenAI,

2024) and DeepSeek R1 (Guo et al., 2025), adopt a strategy of generating intermediate reasoning steps before producing final answers. This approach markedly improves their efficacy in domain-specific tasks, such as mathematical reasoning (Jia et al., 2024; Gao et al., 2024; Lai et al., 2024; Lightman et al., 2023; Huang et al., 2024). The remarkable success of this technology is mainly attributed to the Reinforcement Fine-Tuning (RFT) method (Schulman et al., 2017; Shao et al., 2024; Yu et al., 2025; Li et al., 2024; Hu, 2025; Dai et al., 2026; Ji et al., 2026a; Li et al., 2026; Xiong et al., 2025). Through the application of RFT, the models allocate additional time to "deliberate" prior to generating answers, thereby constructing intricate reasoning chains and subsequently enhancing overall model performance.

In contrast to Supervised Fine-Tuning (SFT), which involves training models on fixed input-output pairs to mimic correct responses, RFT introduces an iterative process that incentivizes models to generate coherent and logically structured reasoning paths. RFT leverages RL techniques, such as Proximal Policy Optimization (PPO) (Schulman et al., 2017) and GRPO (Shao et al., 2024) to optimize decision-making during the generation of intermediate steps. Specifically, PPO ensures stability by constraining policy updates, preventing new strategies that deviate significantly from established behaviours. In contrast, GRPO enhances this process by evaluating performance across groups of actions, encouraging consistent improvements in reasoning quality. This dynamic and feedback-driven approach enables models to think more deeply, resulting in nuanced answers that better handle complex reasoning tasks compared to the more rigid and label-dependent training of SFT.

Despite the significant success of PPO in enhancing reasoning quality, it still suffers severely from the enormous resource consumption required during training. PPO necessitates the development and integration of both a critic model and a reference model, which not only complicates the training process but also substantially increases computational demands. Consequently, there is a growing trend toward simplifying the PPO method. For instance, ReMax (Li et al., 2024) removes the critic model by introducing a baseline value, which reduces the training GPU memory usage and accelerates the training process. Besides, GRPO eliminates the need for a critic model and utilizes normalized rewards within a sample group.

In addition to these methods to improve efficiency and stability, a very recent and concurrent work Dr. GRPO (Liu et al., 2025a) studies the details of reward and loss normalization and states GRPO tends to generate more tokens. However, although it reveals the reward bias in the advantage function, we observe that its performance did not significantly outperform GRPO.

Let's review the birth of PPO. PPO was proposed as a general RL algorithm, with Atari games as primary evaluation benchmarks, where the policy network typically learns both visual representations and the control policy. In the LLM era, however, the policy is an LLM that already possesses strong representations from pretraining and SFT. Removing unnecessary components is important for scalability, which motivates rethinking simplified RL methods. Notably, PPO itself is a simplification of TRPO (Schulman et al., 2015), which in turn builds on the policy-gradient algorithm. A major weakness of policy gradients is high variance, which can be mitigated by (i) using a value-function baseline in advantage estimation and (ii) sampling more trajectories—both common practices in the post-training training for LLMs. Thus, it is natural to build a streamlined method for reasoning.

In summary, our key contributions are as follows:

- We revisit the design of policy gradient algorithms (Sutton et al., 1998) and propose a simple RL method that retains minimal RL components. Unlike conventional approaches, our method directly optimizes the objective function rather than relying on surrogate loss.
- Our approach eschews the necessity for both a critic model and a reference model. Moreover, it imposes no distributional constraints. These characteristics confer substantial advantages for potential scalability.
- We analyze and demonstrate the reward bias inherent in existing advantage functions and reveal the limitations of simplistic debiasing methods. Our exploration of the gradient estimate bias phenomenon has led us to propose a simple yet accurate gradient estimation (AGE) technique. To mitigate the potential issue of large variance in gradient estimation when the proportion of valid samples is excessively small, we introduce a simple thresholding mechanism to ensure a minimal partition of valid samples is maintained, followed by resampling.

- Extensive experiments demonstrate that GPG achieves SOTA results across various unimodal and multimodal visual tasks.

## 2 METHOD

### 2.1 PRELIMINARY AND TASK FORMULATION

RL is a computational approach to learning through interaction, where an agent seeks to maximize cumulative rewards by selecting optimal actions within an environment. The RL problem is typically defined by a policy $\pi_\theta$, which maps states to actions, and aims to optimize the expected return. The core idea behind policy gradient methods is to use gradient ascent to iteratively adjust the policy parameters. The learning objective is maximizing the return $\mathcal{J}(\theta)$,

$$\mathcal{J}(\theta) = \max_\theta \mathbb{E}_{\pi_\theta} \left[ \sum_{t=0}^{T} r_t \right]. \tag{1}$$

The policy gradient theorem (Sutton et al., 1998) proves that the above problem can be converted into estimating the gradient,

$$\nabla_\theta \mathcal{J}(\theta) = \mathbb{E}_{\pi_\theta} \left[ \nabla_\theta \log \pi_\theta(a_t \mid s_t) Q^{\pi_\theta}(s_t, a_t) \right], \tag{2}$$

where $Q^{\pi_\theta}(s_t, a_t)$ is the action-value function, representing the expected return when taking action $a_t$ in state $s_t$ and following policy $\pi_\theta$ thereafter.

To reduce the variance, the advantage function $A^{\pi_\theta}(s_t, a_t)$ is often used, leading to the policy gradient update rule:

$$\nabla_\theta \mathcal{J}(\theta) = \mathbb{E}_{\pi_\theta} \left[ \nabla_\theta \log \pi_\theta(a_t \mid s_t) A^{\pi_\theta}(s_t, a_t) \right]. \tag{3}$$

*One-step advantage estimation* can be mathematically formulated as (Sutton et al., 1998):

$$A^{\pi_\theta}(s_t, a_t) = Q^{\pi_\theta}(s_t, a_t) - V^{\pi_\theta}(s_t), \tag{4}$$

where $V^{\pi_\theta}(s_t)$ is a function of $s_t$. In principle, $V^{\pi_\theta}(s_t)$ can take any functional form. One commonly employed function is the value function, which represents the expected return when starting from state $s_t$ and following policy $\pi_\theta$. While GAE (Schulman et al., 2018) offers a more sophisticated approach to balance bias and variance in advantage estimation, we find that in the context of model reasoning, one-step estimation is sufficiently effective for achieving good performance. This simplicity is particularly advantageous in scenarios where computational efficiency is paramount.

Given a sequence of questions and instructions, the model is tasked with generating corresponding answers. Subsequently, rewards are returned based on predefined reward models or hand-crafted rules. Our objective is to leverage these reward signals to optimize our policy, thereby enhancing the model's ability to generate accurate and contextually appropriate responses.

However, designing or obtaining accurate rewards for intermediate steps is nontrivial (Guo et al., 2025). To address this challenge, we simplify our problem as follows. Given a question and prompt $s$, we sample an action $a$ from policy $\pi_\theta$ and obtain a final reward signal $r$. Note that the policy distribution $\pi_\theta$ is modeled in an autoregressive manner. In this setting, we can leverage policy gradient methods to optimize the policy.

### 2.2 GROUP POLICY GRADIENT

Our proposed method, Group Policy Gradient (GPG), is designed to address the issue of high variance in policy gradient estimation in the absence of a value model. By leveraging group-level rewards, GPG stabilizes learning and enhances the robustness of reinforcement learning training. Specifically, GPG utilizes the mean reward within each group to normalize the rewards, thereby effectively reducing variance. This approach eliminates the need for a traditional value model, thereby simplifying the training process and enhancing computational efficiency. The name "Group Policy Gradient" reflects our method's core mechanism of utilizing group-level mean rewards to stabilize and optimize learning.

The core objective of GPG is defined as:

$$\mathcal{J}_{\text{GPG}}(\theta) = \mathbb{E}_{(q,a)\sim\mathcal{D},\{o_i\}_{i=1}^{G}} \left[ \frac{1}{\sum_{i=1}^{G} |o_i|} \sum_{i=1}^{G} \sum_{t=1}^{|o_i|} \left( -\log \pi_\theta(o_{i,t} \mid q, o_i, < t)\hat{A}_{i,t} \right) \right], \quad (5)$$

where $o_i$ represents the individual responses in the group $G$, and the advantage of the $i$-th response is calculated by normalizing the group-level rewards $\{R_i\}_{i=1}^{G}$:

$$\hat{A}_{i,t} = \frac{r_i - \text{mean}(\{R_i\}_{i=1}^{G})}{F_{norm}}. \quad (6)$$

$F_{norm}$ is an **optional** normalization technique, which is commonly applied in conjunction with reward clipping to mitigate the impact of unexpected outlier values. One widely adopted practice is to employ standard variance normalization within a training batch (Mnih et al., 2016; Schulman et al., 2017). This approach helps stabilize the training process by reducing the variance of the reward signal, which is particularly important when dealing with environments where the magnitude of rewards can vary significantly, such as in different Atari games. By normalizing the reward signal, the model becomes less sensitive to extreme values, thereby improving the robustness and convergence of the training algorithm. However, in the reasoning tasks involving large models, the reward is typically well-defined and does not suffer from the same variance issues observed in other environments. As for the Math reasoning problem, it is a common practice to award the right answer with 1.0 and the wrong answer with 0.0.

We utilize a basic Math Reasoning setting [1] of SimpleRL from open-r1 (Face, 2025), using only the MATH-lighteval dataset to facilitate rapid experimental validation. Specifically, we remove the format reward and only enable the accuracy reward for simplicity.

| Models | Average | AIME24 | MATH-500 | AMC23 | Minerva | OlympiadBench |
|---|---|---|---|---|---|---|
| Qwen2.5-Math-7B | 30.9 | 13.3 | 57.6 | 45.0 | 14.7 | 23.7 |
| GPRO | 43.7 | 16.7 | 73.4 | 62.5 | 30.2 | 35.7 |
| GPG($F_{norm}$=1,$\alpha$=1) | 43.9 | 23.3 | 76.3 | 52.5 | 30.1 | 37.4 |
| GPG($F_{norm}$=std$\{R(o)\}$,$\alpha$=1) | 45.3 | 23.3 | 73.6 | 60.0 | 30.5 | 39.3 |
| GPG($F_{norm}$=std$\{R(o)\}$,$\alpha = \frac{B}{B-M}$) | 44.1 | 23.3 | 74.2 | 52.5 | 30.9 | 39.7 |
| GPG($F_{norm}$=1, $\alpha = \frac{B}{B-M}$) | 47.8 | 30.0 | 75.0 | 62.5 | 33.1 | 38.2 |
| GPG($F_{norm}$=1, $\alpha = \frac{B}{B-M}, \beta_{th} = 0.6$) | 48.3 | 30.0 | 76.2 | 62.5 | 34.2 | 39.0 |
| Dr. GRPO[†] | 43.7 | 26.7 | 74.6 | 50.0 | 30.1 | 37.3 |

Table 1: Math reasoning results on Qwen2.5-Math-7B model. †: reproduction use the released code.

The critical component: $\hat{A}_{i,t}$, has been underexplored in prior research in reasoning. This gap in the literature highlights the need for further investigation of the role and impact of $\hat{A}_{i,t}$ within reasoning tasks. There are two unresolved problems.

**The $\hat{A}_{i,t}$ should not introduce reward bias**. Otherwise, bias deviates from the original problem formulation. GRPO (Shao et al., 2024) formulates it as $F_{norm} = \text{std}\{R(o)\}$, which is essentially a function of $s_t$ in Equation 2 and explicitly introduces the reward bias. Since we aim to solve the original problem, we don't want to apply a surrogate or bias. However, As shown in Table 1, if we remove this bias item, i.e. $F_{norm} = 1$, it (43.9%) *cannot clearly outperform* GRPO (43.7%), which is opposite to the observation of a concurrent work Dr. GRPO (Liu et al., 2025a).

**Examples of all right or wrong responses within a group introduce bias for the estimation of the gradient**. Given a training batch of batch size $B$, let the gradient of the $i$-th sample be denoted as $g_i$. Without loss of generality, assume that the first $M$ examples within the batch are all right or wrong responses within a group. The standard backpropagation (BP) algorithm estimates the gradient as: $\mathbf{g} = \frac{\sum_{i=1}^{B} \mathbf{g_i}}{B} = \frac{\sum_{i=M+1}^{B} \mathbf{g_i}}{B}$. However, the first $M$ examples are not valid for gradient estimation and contribute zero gradient. Therefore, the more accurate gradient estimation (AGE) can be written as:

$$\hat{\mathbf{g}} = \frac{\sum_{i=M+1}^{B} \mathbf{g_i}}{B - M} = \mathbf{g} \frac{B}{B - M} = \alpha\mathbf{g}, \alpha = \frac{B}{B - M}. \quad (7)$$

---

[1] huggingface/open-r1/recipes/Qwen2.5-Math-7B/grpo/config_simple_rl.yaml

It should be noted that the value $\alpha$ is not a constant and it varies across different sample batches. We also illustrate $\alpha$ with different steps in Figure 2, which indicates the necessity of gradient correction. As for multi-GPU training, to achieve more accurate gradient calculations, it is advisable to gather all non-zero gradient samples across all GPUs and compute the average gradient uniformly. This approach can be implemented through a custom gradient aggregation function, which leads to increased communication overhead. Instead, we derive another equivalent format, which doesn't require extra cost, and we provide the proof in Section A. Therefore, given a batch sample, the objective can be written as

$$\hat{\mathcal{J}}_{\text{GPG}}(\theta) = \alpha \mathcal{J}_{\text{GPG}}(\theta). \tag{8}$$

As shown in Table 1, our method achieves an average score of 47.8%, being equipped with AGE.

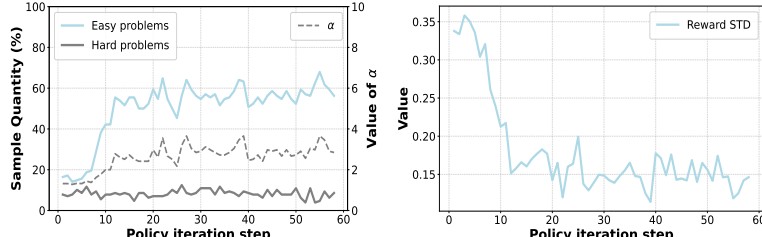

Figure 2: (**Left**) The proportion of easy problems with all rewards are 0, hard problems with all rewards are 1 within a rollout group. (**Right**) The standard deviation of reward across steps.

In a scenario where we reject the $M$ examples and resample responses in a manner similar to the approach presented in a recent work (Yu et al., 2025) until $M$ equals 0, $\alpha$ is set to 1. However, this particular setting is not training-efficient. The reason is that the training time is constrained by the worker that takes the longest to collect the desired examples. In contrast, our proposed method demonstrates superior efficiency. Moreover, it can automatically adjust the loss based on the performance of the sample batch.

We also evaluate a setting of reward normalization of GRPO, where $F_{norm}=\text{std}\{R(o)\}$,$\alpha$=1, and show the result in Table 1. It outperforms $F_{norm}$=1,$\alpha$=1 by 1.4% average score. This motivates us to dive into the source of the improvement. We plot the std of the reward in Figure 2. Note that the std is calculated by averaging the std of each group, whose value ranges from 0.10 to 0.35. And $\alpha$ varies from 1.5 to 4.0. The reward normalization of GRPO provides such a diving std (within a group) mechanism, which has some gradient correction effect.

| | Components | | | |
|---|---|---|---|---|
| | Value Models | Reference Models | Surrogate Loss | Policy Constraint |
| PPO | ✓ | ✓ | ✓ | ✓ |
| GRPO | ✗ | ✓ | ✓ | ✓ |
| TRPO | ✓ | ✗ | ✓ | ✓ |
| GPG | ✗ | ✗ | ✗ | ✗ |

Table 2: Comparison of reinforcement learning algorithms (in reasoning) with various components.

**Thresholding minimal partition of valid samples and resampling to reduce variance.** While our approach provides an unbiased estimation of the gradient, it may encounter issues with high variance when the proportion of valid samples is excessively low. To mitigate this, we introduce a threshold $\beta_{th} = \frac{1}{\alpha_{th}}$ for the proportion of valid samples. When this proportion falls below the given value, we accumulate the valid samples into the resampled subsequent batch until the proportion exceeds the threshold. This strategy effectively reduces the variance of the gradient estimation, thereby enhancing the stability and convergence rate of the model training process. It is worth noting that this strategy further improves the performance, as demonstrated in Table 1.

RL algorithms vary significantly in their approaches to tackling variance and optimizing policies. Two key components in many RL algorithms are surrogate loss and policy constraints. We summarize the main comparisons among various frameworks in Table 2. Our method stands out by preserving

the simplest form, which not only ensures ease of implementation but also maintains high efficiency and effectiveness.

## 3 EXPERIMENTS

All experimental settings are meticulously controlled to ensure fair comparisons. We closely adhere to the hyperparameters employed by GRPO, despite their suboptimality for our approach. Notably, our method consistently outperforms GRPO across all tasks, achieving superior performance with clear margins. These results underscore the robustness and efficacy of our proposed method.

### 3.1 EXPERIMENTAL SETUP

**Dataset and Benchmarks.** In the unimodal scenario, we utilize datasets from multiple sources such as open-s1, open-rs (Dang & Ngo, 2025), and MATH-lighteval (Hendrycks et al., 2021) for training. Specifically, we train the DeepSeek-R1-Distill-Qwen-1.5B base model with the open-s1 dataset, resulting in the GPG-RS1 model. Similarly, training with the open-rs dataset produces the GPG-RS3 model. Furthermore, we perform ablation studies using the MATH-lighteval dataset on the Qwen2.5-Math-7B base model. To compare the overall performance on the 7B model, we utilize the dataset from (Yu et al., 2025), and the detailed setting is shown in Section B.1.

These datasets encompass a wide range of problem types and difficulty levels. To assess the reasoning capabilities of the models, we employ five distinct mathematics-focused benchmark datasets: AIME24, MATH-500 (Lightman et al., 2023; Hendrycks et al., 2021), AMC23, Minerva (Lewkowycz et al., 2022), and OlympiadBench (Huang et al., 2024).

In the multimodal case, we handle a variety of tasks. Specifically, for the visual reasoning task, we utilize approximately $12,000$ samples from the SAT dataset (Ray et al., 2024) for training and perform evaluations on the CV-Bench dataset Tong et al. (2024). In addressing the geometry reasoning task, by following R1-V (Chen et al., 2025), we train on around $8,000$ samples from the GEOQA training set (Chen et al., 2025) and subsequently evaluating performance on the GEOQA test set (Chen et al., 2022). For both the classification and reasoning grounding tasks, we follow Visual-RFT to conduct few-shot classification training on Flower102 (Nilsback & Zisserman, 2008), Pets37 (Parkhi et al., 2012), FGVCAircraft (Maji et al., 2013), Car196 (Krause et al., 2013), respectively. Additionally, training is conducted on 239 samples from the LISA training set (Lai et al., 2024). All evaluations are carried out using the corresponding test sets associated with these training sets.

**Implementation Details.** Our approach is broadly applicable across a wide range of reinforcement learning tasks. To demonstrate its versatility and efficacy, we conduct experiments encompassing both unimodal and multimodal scenarios. These experiments are performed on NVIDIA H20 GPUs and NPUs from China. For each experiment, we adhere strictly to the implementation of original code base, ensuring consistent training and evaluation procedures. The implemented GPG method can refer to Algorithm 1, and more detailed settings can refer to Appendix B.

---

**Algorithm 1** Group Policy Gradient (GPG)

**Input:** $o$ [shape: $(B, G, C, dim)$] ← Model Output, $r$ ← Reward, $\beta_{th}$
  1: Collecting samples and calculate $\hat{A}$ and $\alpha$ based on Equation 6 and Equation 7 until $\alpha < \frac{1}{\beta_{th}}$
  2: Calculate $\log \pi_\theta(o)[per\_token\_logps]$ based on $o$ and model $\pi_\theta$
  3: $loss \leftarrow -\log \pi_\theta(o) \cdot \hat{A} * \alpha$
**Output:** $loss$

---

### 3.2 UNIMODAL TASK EVALUATION

To evaluate our method, we select two models: a strong 1.5B distilled SFT model (DeepSeek-R1-Distill-Qwen-1.5B) and a 7B base model.

**Mathematical Reasoning using 1.5B model (A strong SFT model).** Compared with other 1.5B distilled models, our models exhibit superior performance with average accuracy 55.7% of GPG-RS1, as illustrated in Table 3. Additionally, GPG-RS1 and GPG-RS3 shows strong results in AMC23 with

| Distilled 1.5B Models | Average | AIME24 | MATH-500 | AMC23 | Minerva | OlympiadBench |
|---|---|---|---|---|---|---|
| DeepSeek-R1-Distill-Qwen-1.5B | 48.9 | 28.8 | 82.8 | 62.9 | 26.5 | 43.3 |
| Still-3-1.5B-Preview | 51.6 | 32.5 | 84.4 | 66.7 | 29.0 | 45.4 |
| Open-RS1[†] | 53.1 | 33.3 | 83.8 | 67.5 | 29.8 | 50.9 |
| Open-RS3[†] | 52.0 | 26.7 | 85.4 | 70.0 | 27.9 | 50.2 |
| GPG-RS1 | 55.7 | 33.3 | 87.6 | 77.5 | 29.4 | 50.5 |
| GPG-RS3 | 55.5 | 33.3 | 85.0 | 80.0 | 26.8 | 52.4 |

Table 3: The zero-shot pass@1 performance of the 1.5B models distilled by DeepSeek-R1 across five mathematical reasoning benchmarks. †: reproduced results using released codes. ‡: results from (Dang & Ngo, 2025).

| 7B Models | Average | AIME24 | MATH-500 | AMC23 | Minerva | OlympiadBench |
|---|---|---|---|---|---|---|
| Qwen-2.5-Math-7B-Instruct [‡] | 43.8 | 13.3 | 79.8 | 50.6 | 34.6 | 40.7 |
| Qwen2.5-Math-7B | 30.9 | 13.3 | 57.6 | 45.0 | 14.7 | 23.7 |
| Qwen2.5-Math-7B (no template)[*] | 38.2 | 0.2 | 69.0 | 45.8 | 21.3 | 34.7 |
| rStar-Math-7B (Guan et al., 2025) | - | 26.7 | 78.4 | 47.5 | - | 47.1 |
| Eurus-2-7B-PRIME (Cui et al., 2025) | 48.9 | 26.7 | 79.2 | 57.8 | 38.6 | 42.1 |
| Oat-Zero-7B (Liu et al., 2025a) | 51.4 | 43.3 | 80.0 | 62.7 | 30.1 | 41.0 |
| Oat-Zero-7B (Liu et al., 2025a)[†] | 47.8 | 30.0 | 80.6 | 55.4 | 29.0 | 44.0 |
| OpenReasoner-Zero-7B @ 8k (Hu et al., 2025) | 45.9 | 13.3 | 82.4 | 54.2 | 31.6 | 47.9 |
| SimpleRL-Zero-7B (Zeng et al., 2025)[*] | 46.6 | 26.7 | 78.2 | 60.2 | 27.6 | 40.3 |
| GPG-Zero-7B | 57.7 | 36.7 | 84.6 | 82.5 | 39.0 | 45.8 |

Table 4: The zero-shot pass@1 performance of the 7B models across five mathematical reasoning benchmarks. †: reproduced results using the released code. ‡: results from (Dang & Ngo, 2025), *: results from (Liu et al., 2025a).

a score of 77.5% and 80.0%, obviously surpassing Open-RS 67.5% and 70.0%. Both GPG-RS1 and GPG-RS3 demonstrate competitive performance across various benchmarks, particularly excelling in MATH-500 with scores of 87.6% and 85.0%, and OlympiadBench with scores of 50.5% and 52.4%.

**Mathematical Reasoning using 7B model.** As illustrated in Table 4, GPG-7B achieves an average score of 57.7% and outperforms other baselines with clear margins. This exceptional performance is further highlighted in the AMC23 and Minerva, where GPG-7B attained a leading score of 82.5% and 39.0%, exceeding SimpleRL-Zero-7B by impressive margins of 22.3% and 11.4%, respectively. Moreover, GPG-7B consistently exhibits superiority across most benchmarks, outperforming the recent state-of-the-art method, Oat-Zero-7B, by an average of 6.3%.

| Models | GEOQA$_{Test}$ |
|---|---|
| Qwen2.5-VL-3B-Instruct | 35.41 |
| + GRPO | 47.48 |
| + GPG | 51.33 |

Table 5: Geometry reasoning results on GEOQA. GPG is better than GRPO.

| Models | Average | Flower102 Nilsback & Zisserman (2008) | Pets37 Parkhi et al. (2012) | FGVC Maji et al. (2013) | Cars196 Krause et al. (2013) |
|---|---|---|---|---|---|
| Qwen2-VL-2B | 56.0 | 54.8 | 66.4 | 45.9 | 56.8 |
| + SFT | 55.6 | 58.5 | 55.5 | 67.9 | 40.5 |
| + GRPO | 81.9 | 71.4 | 86.1 | 74.8 | 95.3 |
| + GPG | 89.0 | 79.3 | 90.8 | 88.5 | 97.5 |

Table 6: 4-shot Results on Four Fine-grained Classification Datasets. GPG shows consistently better results than GRPO on 4 classification datasets.

| Models | Total | Count | Relation | Depth | Distance |
|---|---|---|---|---|---|
| Qwen2-VL-2B | 31.38 | 54.69 | 22.46 | 0.16 | 31.66 |
| + SFT | 57.84 | 60.02 | 68.92 | 55.00 | 45.83 |
| + GRPO | 59.47 | 59.64 | 66.76 | 54.16 | 56.66 |
| + GPG | 76.15 | 66.62 | 83.23 | 81.66 | 75.50 |

Table 7: Visual reasoning results on CV-Bench (Tong et al., 2024), which shows GPG training on base model has overall better performance over GRPO and the base model.

| Models | mIoU$_{test}$ | mIoU$_{val}$ | gIoU$_{test}$ |
|---|---|---|---|
| Qwen2-VL-2B | 26.9 | 30.1 | 25.3 |
| + SFT | 28.3 | 29.7 | 25.3 |
| + GRPO | 37.6 | 34.4 | 34.4 |
| + GPG | 51.8 | 51.3 | 50.4 |

Table 8: Reasoning grounding results on LISA (Lai et al., 2024). GPG surpasses GRPO in reasoning grounding.

## 3.3 Multimodal Task Evalutaion

We further evaluate our method on several very recent multimodal benchmarks, most of which report results based on GRPO.

**Geometry Reasoning.** In addition to visual reasoning, MLLMs exhibit notable proficiency in geometry reasoning. To evaluate the efficacy of the GPG method in this domain, we employ an experimental setup similar to that used in R1-V (Chen et al., 2025) using the GEOQA (Chen et al., 2022) dataset. The results, presented in Table 5, indicate that the GPG method achieved a score of 51.33%, surpassing the GRPO's score of 47.48% by 3.85% points. This demonstrates the superior performance of the GPG method in addressing complex geometric reasoning tasks.

**Classification.** Beyond the evaluation of reasoning tasks, we also assess the enhancement of the GPG method over GRPO in image perception tasks. As shown in Table 6, the GPG method achieves an average score of 89.0% across four classification datasets, surpassing GRPO by 7.1% points. Additionally, our method consistently produces improvements across all four classification datasets, underscoring its superiority in image perception tasks.

**Visual Reasoning.** We initially evaluate the GPG method using the CV-Bench (Tong et al., 2024) visual reasoning dataset, strictly adhering to the parameter settings of VisualThinker-R1-Zero. As illustrated in Table 7, the GPG method demonstrates a significant improvement in performance. Specifically, it attains a score of 76.15% on CV-Bench, representing an increase of 16.68% points compared to the 59.47% score achieved by GRPO.

**Reasoning Grounding.** The final critical aspect of evaluating MLLMs involves precisely identifying objects according to user requirements. To this end, we employ the Qwen2-VL-2B model for grounding tasks using the LISA dataset (Lai et al., 2024), with the results presented in Table 8. In comparison to the GRPO method, the GPG approach demonstrates a substantial enhancement, improving all metrics by over 14.0% points. This significant improvement underscores the superiority of the GPG method in object localization, leading to considerable advancements in reasoning and perception capabilities.

## 3.4 Ablation Study and Discussion

**Case Study and Training Analysis.** We present the reasoning processes of GPG and GRPO, as illustrated in Figure 4 (supplementary). Compared to GRPO, the GPG approach demonstrates a more comprehensive and accurate reasoning capability, whereas GRPO exhibits errors in formula analysis. Consequently, GPG arrives at the correct solution, while GRPO produces an incorrect result. In Figure 3, we present a range of real-time training metrics to illustrate the effectiveness of GPG as a straightforward yet strong RL algorithm.

**Sensitivity on Group Size.** We study the effect of the number of generations within a group. As shown in Table 11, increasing the group size from 2 to 16 leads to progressive improvements across most metrics. Specifically, the Average performance improves steadily with larger group sizes. We choose 8 to achieve a good tradeoff between training cost and performance.

**Comparison with Various RL Methods.** We attempt to explain the differences between GPG and other RL methods in the simplest way. As shown in Table 14, it can be seen that the loss of GPG does not include the "CLIP term" and the "KL divergence". Its form and calculation are the simplest, and as discussed in Section 3.2, its performance is better than other methods.

**Comparison with DAPO (Yu et al., 2025).** We meticulously control the experimental settings and rigorously reported the results in Table 9. All models are trained on the same dataset and for the same number of steps (1100). In contrast to DAPO (Yu et al., 2025), which incorporates all proposed components, our method focuses exclusively on the accuracy reward. Despite this, our approach achieves superior performance with reduced training and data costs. DAPO, which constructs fully valid batches through dynamic sampling, often requires more batches and may waste valid samples in the final batch. In contrast, our method avoids these inefficiencies, ensuring optimal resource utilization and enhanced performance.

**KL constraint.** In principle, our method is designed to optimize the original reinforcement learning (RL) problem directly. And it's a bit strange without imposing any distribution constraints. Despite

| Method | Average | AIME24 | MATH-500 | AMC23 | Minerva | OlympiadBench | Training Cost | Data Cost | Memory |
|--------|---------|--------|----------|-------|---------|---------------|---------------|-----------|--------|
| DAPO-7B | 56.0 | 30.0 | 84.6 | 82.5 | 34.9 | 47.8 | $1\times$ | $1\times$ | 28G |
| GPG-Zero-7B | 57.7 | 36.7 | 84.6 | 82.5 | 39.0 | 45.8 | $0.45\times$ | $0.39\times$ | 24G |

Table 9: Comparison with DAPO (Qwen-7B Math). Ours is simpler, stronger and resource efficient.

this, we conduct an ablation study to evaluate the impact of adding a distribution constraint. The results are presented in Table 13. Our findings indicate that incorporating such a constraint negatively impacts performance.

Limited by space, we provide more ablation studies in Section B.2.

### 3.5 IMPACT AND LIMITATION DISCUSSION

Achieving advanced general intelligence critically depends on augmenting the reasoning capabilities of models, with efficient and scalable reinforcement learning methods serving as a cornerstone. Our proposed approach investigates a minimalist strategy that aims to enhance reasoning capacity through simplicity and efficiency, thereby potentially facilitating the development of scalable systems. However, given the constraints of our computational budget, we do not evaluate our method on extremely large models.

## 4 RELATED WORK

**Large Model Reasoning.** Recent advancements in both LLM and Multimodal Large Language Model (MLLM) increasingly focus on enabling models to simulate human-like, stepwise reasoning processes. In the field of LLMs, researchers have pioneered methods such as Chain-of-Thought (CoT) prompting (OpenAI, 2024; Wei et al., 2022; Kojima et al., 2022; Ye et al., 2025), Tree-of-Thought (Yao et al., 2023), Monte Carlo Tree Search (Feng et al., 2024; Xin et al., 2024; Trinh et al., 2024), and the construction of complex SFT datasets (Muennighoff et al., 2025), to enhance performance in reasoning tasks. Notably, approaches such as DeepSeek-R1 (Guo et al., 2025) employ large-scale RL with format-specific and result-oriented reward functions, guiding LLMs toward self-emerging, human-like, complex CoT reasoning with significant performance improvements in challenging reasoning tasks. Meanwhile, MLLMs convert inputs from various modalities into a unified LLM vocabulary representation space for processing and exhibit superior performance in vision understanding tasks (Wu et al., 2024; Liu et al., 2023; Chen et al., 2024; Google, 2023). Building on advancements in LLM reasoning, the research community collectively applies the DeepSeek-R1 methodology to MLLMs to enhance their visual reasoning capabilities, yielding remarkable progress (Zhang et al., 2025; Liu et al., 2025b; Chen et al., 2025; Zhou et al., 2025; Yuan et al., 2026b; Ji et al., 2026b; Wang et al., 2026; Yuan et al., 2026a).

**Reinforcement Learning.** RL has driven significant progress in sequential decision-making, with policy gradient methods being fundamental to optimizing stochastic policies. The REINFORCE algorithm (Williams, 1992) establishes early principles for gradient-based policy updates in trajectory-driven tasks. However, its high variance poses challenges for scalability. To address this, subsequent research focus on stabilizing policy optimization processes. Trust Region Policy Optimization (TRPO) (Schulman et al., 2015) introduces constrained updates via quadratic approximations to ensure monotonic improvement. This approach is further refined by PPO (Schulman et al., 2017), which employed clipped objective functions to simplify the optimization process. Subsequent studies seek to enhance the PPO algorithm (Zheng et al., 2023) or elaborate on its implementation (Engstrom et al., 2019). PPO achieves widespread use in language model alignment and robotic control. However, the algorithm's dependence on conservative policy updates or heuristic clipping thresholds can undermine its exploration potential in favour of stability, which poses a significant challenge in complex domains requiring dynamic strategy adaptation.

Limited by space, more related work is discussed in Section C.

## 5 CONCLUSION

In this paper, we introduce GPG, which effectively addresses the critical challenges inherent in reinforcement fine-tuning approaches such as PPO and GRPO. By directly incorporating group-based decision dynamics into the standard PG method, GPG simplifies the training process and significantly reduces computational overhead without sacrificing reasoning quality. This breakthrough provides a more efficient framework for training advanced LLMs capable of complex reasoning, thereby contributing to more resource-effective and scalable artificial intelligence systems.

## 6 REPRODUCIBILITY STATEMENT

We have taken the following steps to ensure the reproducibility of our empirical results: (1) We provide a comprehensive description of all experimental setups, including the datasets used and their corresponding benchmarks, in Section 3.1. (2) Our implementation builds upon several publicly available code repositories. For unimodal tasks, we utilize the VERL framework (Sheng et al., 2024), Open-r1 (Face, 2025), and Open-rs (Dang & Ngo, 2025). For multimodal tasks, we adopt VisualThinker-R1-Zero (Zhou et al., 2025), R1-V (Chen et al., 2025), and Visual-RFT (Liu et al., 2025b). These repositories have been adapted to suit our purposes and to facilitate replication by the research community. (3) Detailed training configurations—including hyperparameters, evaluation protocols, and specific adaptations applied to each base framework—are thoroughly documented in Appendices B.1. (4) In compliance with the double-blind review policy, we have made our full implementation, along with training and evaluation scripts, publicly accessible through an anonymous repository. This ensures that all reported results can be reproduced without revealing the authors' identities.

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

## A ANALYSIS OF DISTRIBUTED GRADIENT AVERAGING WITH INVALID SAMPLES

### A.1 PROBLEM FORMULATION

Consider a distributed training setup where:

- A batch of $B$ samples is evenly distributed across $N$ GPUs, with each GPU processing $K = B/N$ samples.
- For the $i$-th GPU, the first $M_i$ samples produce zero gradients (invalid samples), while the remaining $(K - M_i)$ samples generate valid gradients.
- Let $M_{\text{total}} = \sum_{i=1}^{N} M_i$ denote the total invalid samples, and $S = B - M_{\text{total}}$ the effective valid samples.

Let $g_{i,j}$ represent the gradient of the $j$-th valid sample on the $i$-th GPU. We define the valid gradient sum for GPU $i$ as:

$$G_i = \sum_{j=M_i+1}^{K} g_{i,j} \tag{9}$$

The conventional distributed averaging approach in PyTorch computes a gradient estimate:

$$\hat{G}_{\text{PyTorch}} = \frac{1}{B} \sum_{i=1}^{N} G_i \tag{10}$$

whereas the theoretically correct gradient should be:

$$G_{\text{true}} = \frac{1}{S} \sum_{i=1}^{N} G_i \tag{11}$$

*Proof.* **Step 1: Conventional Approach Derivation**

Each GPU calculates its local mean using the *assigned* sample count $K$ (not valid samples):

$$\bar{G}_i^{\text{local}} = \frac{G_i}{K} \tag{12}$$

Global averaging then gives:

$$\hat{G}_{\text{PyTorch}} = \frac{1}{N} \sum_{i=1}^{N} \bar{G}_i^{\text{local}} = \frac{1}{N} \sum_{i=1}^{N} \frac{G_i}{K} = \frac{1}{B} \sum_{i=1}^{N} G_i \quad (\text{since } N \cdot K = B) \tag{13}$$

**Step 2: True Gradient Computation**

The correct gradient averages over only valid samples:

$$G_{\text{true}} = \frac{1}{S} \sum_{i=1}^{N} G_i \quad (S = B - M_{\text{total}}) \tag{14}$$

Observe the proportional relationship:

$$\hat{G}_{\text{PyTorch}} = \frac{1}{B} \sum_{i=1}^{N} G_i \tag{15}$$

$$= \left( \frac{1}{S} \sum_{i=1}^{N} G_i \right) \cdot \frac{S}{B} \tag{16}$$

$$= G_{\text{true}} \cdot \frac{S}{B} \tag{17}$$

$\square$

## B MORE EXPERIMENT DETAILS

### B.1 EXPERIMENT SETTINGS

**Training setting on 7B based on dataset from (Yu et al., 2025).** We employ the VERL framework (Sheng et al., 2024) with a global batch size of 144 prompts. For each prompt, we generate 8 responses and use only accuracy-based rewards. Our implementation strictly follows Algorithm 1. We optimize the network using the AdamW optimizer with a constant learning rate of $1 \times 10^{-6}$ and a weight decay of 0.1. The threshold value $\beta_{th}$ is set to 0.6. We trained the model for 1100 steps utilizing 48 NPUs sourced from China.

To evaluate the unimodal reasoning capabilities of our proposed method, we utilize two publicly available code repositories: Open-r1 (Face, 2025) and Open-rs (Dang & Ngo, 2025). These repositories are selected due to their extensive coverage of various reasoning scenarios and their ability to present substantial challenges that effectively assess the reasoning capabilities of advanced models. The DeepSeek-R1-Distill-Qwen-1.5B model is trained for 100 and 50 global steps using the open-s1 and open-rs datasets, as reported in the repository (Dang & Ngo, 2025), resulting in the GPG-RS1 and GPG-RS3 models, respectively.

For multimodal tasks, we have selected three renowned frameworks as our code base: VisualThinker-R1-Zero (Zhou et al., 2025), R1-V (Chen et al., 2025), and Visual-RFT (Liu et al., 2025b). These frameworks cover a variety of tasks, including visual reasoning, geometric reasoning, and image perception. The use of distinct code bases enables a comprehensive assessment of the performance enhancements achieved by our method across different tasks. Specifically, for the VisualThinker-R1-Zero framework, we evaluate the results of the GPG approach on the CV-Bench (Tong et al., 2024). Additionally, we evaluate our method on the GEOQA dataset (Chen et al., 2022) based on R1-V. Finally, for tasks related to image perception, such as classification (Nilsback & Zisserman, 2008; Parkhi et al., 2012; Maji et al., 2013; Krause et al., 2013) and reasoning grounding (Lai et al., 2024), we examine the performance of GPG using the Visual-RFT framework.

### B.2 MORE ABLATION EXPERIMENT RESULTS

| $\beta_{th}$ | Average | AIME24 | MATH-500 | AMC23 | Minerva | OlympiadBench |
|---|---|---|---|---|---|---|
| 0.6 | 48.3 | 30.0 | 76.2 | 62.5 | 34.2 | 39.0 |
| 0.8 | 48.6 | 33.3 | 73.6 | 67.5 | 29.4 | 39.3 |

Table 10: Ablation on different $\beta_{th}$ using Qwen2.5 Math 7B.

| Group Number | Average | AIME24 | MATH-500 | AMC23 | Minerva | OlympiadBench |
|---|---|---|---|---|---|---|
| 2 | 41.9 | 16.7 | 71.6 | 60.0 | 25.0 | 36.0 |
| 4 | 43.3 | 20.0 | 73.2 | 55.0 | 29.8 | 38.5 |
| 8 | 45.3 | 23.3 | 73.6 | 60.0 | 30.5 | 39.3 |
| 16 | 47.3 | 26.7 | 74.6 | 65.0 | 32.4 | 37.8 |

Table 11: Ablation on different group size (wo AGE) using Qwen2.5 Math 7B.

| Model | MMLU | C-Eval |
|---|---|---|
| DeepSeek-R1-distill-qwen-1.5B | 38.31 | 32.91 |
| + GPG | 38.53 (+0.22) | 33.29 (+0.38) |

Table 12: Evaluation of GPG on MMLU and C-Eval.

**Reward Normalization.** We study the role of reward normalization and show the result in Table 13. Normalization within a batch is common practice in the RL training process (Andrychowicz et al., 2021). The results of the experiment show that reward normalization within a group is better than the batch.

**Comparision of various RL methods.** We compare the main component of various RL methods in Table 14 and illustrate the evolution from GPRO to GPG in Table 15 .

| $F_{norm}$ | Average | AIME24 | MATH-500 | AMC23 | Minerva | OlympiadBench |
|---|---|---|---|---|---|---|
| Group | 45.3 | 23.3 | 73.6 | 60.0 | 30.5 | 39.3 |
| Batch | 44.9 | 23.3 | 72.2 | 55.0 | 35.3 | 38.5 |
| 1 | 43.9 | 23.3 | 76.3 | 52.5 | 30.1 | 37.4 |

Table 13: Ablation on reward normalization using Qwen2.5 Math 7B.

| RL Method | Loss Function | Advantage Function |
|---|---|---|
| PPO (Schulman et al., 2017) | $\mathcal{L}_{\text{PPO}} = -\min\left[ \frac{\pi_\theta(o)}{\pi_{\theta_{old}}(o)} \cdot A, \underbrace{\text{clip}\left( \frac{\pi_\theta(o)}{\pi_{\theta_{old}}(o)}, 1-\epsilon, 1+\epsilon \right)}_{\text{CLIP}} \cdot A \right]$ | where $A$ computed by applying GAE (Schulman et al., 2018) based on rewards and the **critic model**. |
| GRPO (Shao et al., 2024) | $\mathcal{L}_{\text{GRPO}} = -\left( \min\left[ \frac{\pi_\theta(o)}{\pi_{\theta_{old}}(o)} \cdot A, \text{CLIP} \cdot A \right] - \beta \mathbb{D}_{KL}\left[ \pi_\theta \| \pi_{ref} \right] \right)$ | $A = \frac{R(o) - \text{mean}\{R(o)\}}{\text{std}\{R(o)\}}$ |
| Dr. GRPO (Liu et al., 2025a) | $\mathcal{L}_{\text{Dr.GRPO}} = \mathcal{L}_{\text{PPO}}$ | $A = R(o) - \text{mean}\{R(o)\}$ |
| DAPO (Yu et al., 2025) | $\mathcal{L}_{\text{DAPO}} = -\min\left[ \frac{\pi_\theta(o)}{\pi_{\theta_{old}}(o)} \cdot A, \text{clip}\left( \frac{\pi_\theta(o)}{\pi_{\theta_{old}}(o)}, 1-\epsilon_{\text{low}}, 1+\epsilon_{\text{high}} \right) \cdot A \right]$ | $A = \frac{R(o) - \text{mean}\{R(o)\}}{\text{std}\{R(o)\}}$ |
| **GPG** | $\mathcal{L}_{\text{GPG}} = -\log\pi_\theta(o) \cdot A$ | $A = \alpha * (R(o) - \text{mean}\{R(o)\})$ |

Table 14: Comparison of various RL methods, we explain in the simplest form.

Compared with the GRPO baseline, Group A replaces the loss with a policy-gradient (PG) loss and removes the KL divergence term. It thus applies a policy-gradient algorithm with group rewards, as in GRPO. Group B corrects an error in reward normalization and uses the proper formula; however, its performance degrades. We attribute this degradation to gradient bias, which Group C mitigates via $\alpha$-scaling, yielding improved performance. Group D further improves performance by imposing a minimum valid-sample proportion threshold, which serves as a variance-reduction mechanism.

| Models | Average | Value Models | Reference Models | Surrogate Loss | Policy Constraint | Debiased Gradient | Variance Reduction |
|---|---|---|---|---|---|---|---|
| Qwen2.5-Math-7B | 30.9 | - | - | - | - | - | - |
| GPRO | 43.7 | ✗ | ✓ | ✓ | ✓ | ✗ | ✗ |
| A. GPG($F_{norm} = \text{std}\{R(o)\}, \alpha = 1$) [PG+Group Reward] | 45.3 | ✗ | ✗ | ✗ | ✗ | ✗ | ✗ |
| B. GPG($F_{norm} = 1, \alpha = 1$) | 43.9 | ✗ | ✗ | ✗ | ✗ | ✗ | ✗ |
| C. GPG($F_{norm} = 1, \alpha = \frac{B}{B-M}$) | 47.8 | ✗ | ✗ | ✗ | ✗ | ✓ | ✗ |
| D. GPG($F_{norm} = 1, \alpha = \frac{B}{B-M}, \beta_{th} = 0.6$) | 48.3 | ✗ | ✗ | ✗ | ✗ | ✓ | ✓ |

Table 15: Math reasoning results on Qwen2.5-Math-7B model.

**Robust evaluation across multiple runs.** To more robustly evaluate our method, we additionally report the mean and standard deviation in Table 16. Our method consistently shows clear advantages over other baselines, consistent with the trends in Table 4.

## B.3 EVALUATION ON GENERAL BENCHMARKS

One potential concern for GPG is that the performance gains on specialized reasoning benchmarks might come at the cost of degrading the model's general capabilities. To investigate this, we conduct an additional evaluation on two widely used general benchmarks that are unrelated to the reasoning datasets used in training: MMLU (Hendrycks et al., 2021), which covers 57 subjects spanning STEM, humanities, social sciences, and other fields, and C-Eval (Huang et al., 2023), a comprehensive Chinese evaluation suite consisting of 52 diverse disciplines.

We evaluate the DeepSeek-R1-distill-Qwen-1.5B and the same model after being trained by GPG. The evaluation is performed using the OPENCOMPASS framework, ensuring identical settings for a fair comparison. As shown in Table 17 and 18, GPG achieves consistent improvements on both MMLU (**+0.22**) and C-Eval (**+0.38**), indicating that it not only boosts reasoning-specific benchmarks but also enhances performance on general-purpose evaluations. Detailed results for each sub-domain are provided in Table 17 and Table 18. These findings confirm that GPG's improvements on specialized reasoning tasks do not compromise the model's general capabilities, and in some cases even slightly enhance them. Therefore, GPG can be regarded as a safe and broadly applicable method.

We also report zero-shot evaluation results on code generation and general QA tasks in Table 19, where our method still outperforms GRPO.

| Model | Avg | AIME24 | AMC23 | MATH_500 | MINERVA |
|---|---|---|---|---|---|
| *pass@1 (Acc / Std)* | | | | | |
| Oat-Zero | 52.0 | 31.6/8.8 | 66.5/7.6 | 79.5/1.8 | 30.5/2.8 |
| Eurus-2-7B-PRIME | 48.9 | 16.7/6.8 | 62.5/7.5 | 79.6/1.8 | 37.1/2.9 |
| Open-Reasoner-Zero-7B | 46.3 | 15.8/6.3 | 55.0/7.9 | 82.2/1.7 | 32.2/2.8 |
| Qwen-2.5-Math-7B-SimpleRL-Zero | 49.4 | 29.2/8.5 | 60.6/7.8 | 76.6/1.8 | 31.3/2.9 |
| GPG-Zero-7B | 58.7 | 31.7/8.8 | 80.6/6.1 | 85.3/1.6 | 37.4/3.0 |
| *pass@3 (Acc / Std)* | | | | | |
| Oat-Zero | 59.4 | 40.0/8.8 | 74.4/6.7 | 85.6/1.6 | 37.4/2.9 |
| Eurus-2-7B-PRIME | 58.7 | 27.5/7.4 | 76.3/6.9 | 86.9/1.5 | 44.3/3.0 |
| Open-Reasoner-Zero-7B | 54.8 | 20.8/6.9 | 68.8/7.8 | 88.1/1.5 | 41.5/3.0 |
| Qwen-2.5-Math-7B-SimpleRL-Zero | 58.9 | 36.8/9.1 | 70.0/6.7 | 86.0/1.3 | 42.8/3.0 |
| GPG-Zero-7B | 64.7 | 41.5/9.2 | 85.0/5.7 | 89.0/1.4 | 43.4/3.0 |
| *pass@5 (Acc / Std)* | | | | | |
| Oat-Zero | 62.2 | 42.5/8.9 | 78.8/6.7 | 87.1/1.5 | 40.5/3.0 |
| Eurus-2-7B-PRIME | 62.1 | 30.9/8.2 | 80.6/6.7 | 89.3/1.3 | 47.6/3.0 |
| Open-Reasoner-Zero-7B | 60.4 | 27.5/7.4 | 78.1/6.9 | 90.5/1.3 | 45.3/3.0 |
| Qwen-2.5-Math-7B-SimpleRL-Zero | 62.7 | 41.6/9.1 | 74.4/6.7 | 88.8/1.3 | 45.9/3.0 |
| GPG-Zero-7B | 66.2 | 41.7/9.2 | 86.3/5.7 | 90.9/1.3 | 46.0/3.0 |

Table 16: Pass@3 and pass@5 results—reported as mean ± standard deviation computed over four random seeds—for the Qwen2.5-7B base model. GPG consistently shows clear advantages over other baselines, consistent with the trends in Table 4.

## B.4 PROMPT AND REWARD FUNCTION

**Prompt for Reasoning.** In the process of reinforcement fine-tuning, specific instructions are incorporated into the system prompt. These instructions encourage the model to generate intermediate reasoning steps, thereby facilitating the reasoning capabilities of the model. An example of this approach is provided below (Liu et al., 2025b):

> **System Prompt for Reasoning for 1.5B Model**
>
> A conversation between User and Assistant. The user asks a question, and the Assistant solves it. The assistant first thinks about the reasoning process in the mind and then provides the user with the answer. The reasoning process and answer are enclosed within <think> </think> and <answer> </answer> tags, respectively, i.e., <think> reasoning process here </think><answer> answer here </answer>

> **System Prompt for Qwen 7B Reasoning**
>
> <|im_start|>system\nYou are a helpful assistant.<|im_end|>\n<|im_start|>user\nKelly can read five pages of her fiction book or two pages of her history textbook in seven minutes. If Kelly wants to read thirty pages of each book, for how many minutes in total must Kelly read?\nPlease reason step by step, and put your final answer within boxed.<|im_end|>\n<|im_start|>assistant\n"

**Reward Function.** For most tasks, we use the accuracy and formatting reward functions. For the grounding task, the Intersection over Union (IoU) reward function is utilized. For the Qwen 7B setting, we only use the accuracy reward.

- Accuracy: If the model's output is consistent with the ground truth, a reward of 1.0 is awarded.
- Formatting: If the format of the model output is "<think></think> <answer></answer>", a reward of 1.0 is granted.

| MMLU Datasets | Deepseek-R1-Distill-Qwen-1.5B | +GPG | Accuracy Gain |
|---|---|---|---|
| college biology | 24.31 | 24.31 | 0.00 |
| college chemistry | 32.00 | 32.00 | 0.00 |
| college computer science | 21.00 | 21.00 | 0.00 |
| college mathematics | 32.00 | 32.00 | 0.00 |
| college physics | 28.43 | 30.39 | +1.96 |
| electrical engineering | 50.34 | 50.34 | 0.00 |
| astronomy | 36.84 | 38.16 | +1.32 |
| anatomy | 35.56 | 35.56 | 0.00 |
| abstract algebra | 27.00 | 27.00 | 0.00 |
| machine learning | 37.50 | 37.50 | 0.00 |
| clinical knowledge | 41.89 | 42.64 | +0.75 |
| global facts | 31.00 | 30.00 | -1.00 |
| management | 43.69 | 43.69 | 0.00 |
| nutrition | 38.89 | 38.89 | 0.00 |
| marketing | 58.55 | 58.55 | 0.00 |
| professional accounting | 29.08 | 28.72 | -0.36 |
| high school geography | 44.95 | 45.45 | +0.50 |
| international law | 45.45 | 47.11 | +1.66 |
| moral scenarios | 24.13 | 24.36 | +0.23 |
| computer security | 39.00 | 39.00 | 0.00 |
| high school microeconomics | 44.96 | 46.22 | +1.26 |
| professional law | 27.71 | 28.16 | +0.45 |
| medical genetics | 46.00 | 46.00 | 0.00 |
| professional psychology | 33.5 | 33.66 | +0.16 |
| jurisprudence | 39.81 | 39.81 | 0.00 |
| world religions | 33.92 | 33.33 | -0.59 |
| philosophy | 41.48 | 42.12 | +0.64 |
| virology | 40.96 | 40.96 | 0.00 |
| high school chemistry | 38.42 | 39.41 | +0.99 |
| public relations | 42.73 | 42.73 | 0.00 |
| high school macroeconomics | 42.82 | 43.08 | +0.26 |
| human sexuality | 48.85 | 48.85 | 0.00 |
| elementary mathematics | 37.57 | 38.10 | +0.53 |
| high school physics | 24.50 | 23.84 | -0.66 |
| high school computer science | 42.00 | 42.00 | 0.00 |
| high school european history | 40.00 | 40.61 | +0.61 |
| business ethics | 43.00 | 43.00 | 0.00 |
| moral disputes | 37.57 | 37.28 | -0.29 |
| high school statistics | 50.00 | 50.46 | +0.46 |
| miscellaneous | 44.32 | 44.57 | +0.25 |
| formal logic | 29.37 | 29.37 | 0.00 |
| high school government and politics | 38.34 | 38.86 | +0.52 |
| prehistory | 32.72 | 33.33 | +0.61 |
| security studies | 43.67 | 43.67 | 0.00 |
| high school biology | 44.52 | 44.19 | -0.33 |
| logical fallacies | 38.04 | 38.04 | 0.00 |
| high school world history | 42.62 | 43.04 | +0.42 |
| professional medicine | 38.60 | 38.97 | +0.37 |
| high school mathematics | 30.00 | 30.74 | +0.74 |
| college medicine | 32.37 | 32.95 | +0.58 |
| high school us history | 35.78 | 35.78 | 0.00 |
| sociology | 47.76 | 48.26 | +0.50 |
| econometrics | 32.46 | 32.46 | 0.00 |
| high school psychology | 42.94 | 42.75 | -0.19 |
| human aging | 36.32 | 36.32 | 0.00 |
| us foreign policy | 56.00 | 57.00 | +1.00 |
| conceptual physics | 40.43 | 39.57 | -0.86 |
| **AVERAGE** | **38.31** | **38.53** | **+0.22** |

Table 17: Comparison of performance metrics across general MMLU datasets.

• IoU: Consistent with Visual-RFT (Liu et al., 2025b), the reward value is derived from the calculated scores of the bounding boxes generated by the model.

| C-Eval Datasets | Deepseek-R1-Distill-Qwen-1.5B | GPG | Accuracy Gain |
|---|---|---|---|
| computer network | 9.09 | 9.09 | 0.00 |
| operating system | 42.86 | 43.36 | +0.50 |
| computer architecture | 28.57 | 27.93 | -0.64 |
| college programming | 50.00 | 50.00 | 0.00 |
| college physics | 16.67 | 17.49 | +0.82 |
| college chemistry | 42.86 | 42.86 | 0.00 |
| advanced mathematics | 42.11 | 41.83 | -0.28 |
| probability and statistics | 38.89 | 39.30 | +0.41 |
| discrete mathematics | 20.57 | 21.05 | +0.48 |
| electrical engineer | 38.89 | 38.89 | 0.00 |
| metrology engineer | 12.00 | 13.40 | +1.40 |
| high school mathematics | 20.78 | 21.67 | +0.89 |
| high school physics | 47.18 | 50.67 | +3.49 |
| high school chemistry | 49.00 | 50.25 | +1.25 |
| high school biology | 22.22 | 22.54 | +0.32 |
| middle school mathematics | 26.67 | 26.34 | -0.33 |
| middle school biology | 39.00 | 38.37 | -0.63 |
| middle school physics | 42.86 | 45.93 | +3.07 |
| middle school chemistry | 33.33 | 32.47 | -0.86 |
| veterinary medicine | 46.15 | 47.13 | +0.98 |
| college economics | 57.89 | 57.89 | 0.00 |
| business administration | 15.38 | 19.88 | +4.50 |
| marxism | 30.82 | 30.82 | +0.00 |
| mao zedong thought | 16.67 | 17.06 | +0.39 |
| education science | 45.45 | 45.16 | -0.29 |
| teacher qualification | 55.56 | 57.68 | +2.12 |
| high school politics | 21.43 | 29.24 | +7.81 |
| high school geography | 27.27 | 28.11 | +0.84 |
| middle school politics | 29.56 | 29.56 | 0.00 |
| middle school geography | 4.73 | 4.73 | 0.00 |
| modern chinese history | 25.00 | 25.00 | 0.00 |
| ideological and moral cultivation | 39.98 | 39.98 | 0.00 |
| logic | 46.67 | 46.02 | -0.65 |
| law | 26.67 | 20.33 | -6.34 |
| chinese language and literature | 15.38 | 14.79 | -0.59 |
| art studies | 35.71 | 35.71 | 0.00 |
| professional tour guide | 20.00 | 20.00 | 0.00 |
| legal professional | 7.14 | 6.66 | -0.48 |
| high school chinese | 41.67 | 41.67 | 0.00 |
| high school history | 41.67 | 42.21 | +0.54 |
| middle school history | 9.09 | 9.09 | 0.00 |
| civil servant | 47.37 | 47.84 | +0.47 |
| sports science | 62.48 | 62.48 | 0.00 |
| plant protection | 33.33 | 34.33 | +1.00 |
| basic medicine | 44.44 | 43.74 | -0.70 |
| clinical medicine | 42.86 | 42.16 | -0.70 |
| urban and rural planner | 57.14 | 61.44 | +4.30 |
| accountant | 23.53 | 22.77 | -0.76 |
| fire engineer | 41.67 | 41.67 | 0.00 |
| environmental impact engineer | 19.05 | 18.34 | -0.71 |
| tax accountant | 30.75 | 29.22 | -1.53 |
| physician | 25.00 | 24.79 | -0.21 |
| **AVERAGE** | **32.91** | **33.29** | **+0.38** |

Table 18: Comparison of performance metrics across general C-Eval datasets.

| Method | Code | | General QA | |
|--------|------|------|------------------------------------|------------------------------|
| | MBPP | MBPP+ | HellaSwag ($\text{acc}_{\text{norm}}/std_{err}$) | TruthfulQA ($\text{mc2}/std_{err}$) |
| GRPO | 24.60% | 21.96% | 41.914/0.492 | 47.307/1.516 |
| GPG | 26.19% | 23.81% | 42.551/0.493 | 50.146/1.533 |

Table 19: Zero-shot results on code generation and general QA tasks using Qwen-1.5B.

## B.5 ZERO-SHOT EVALUATION ON GENERAL QA AND CODING TASK

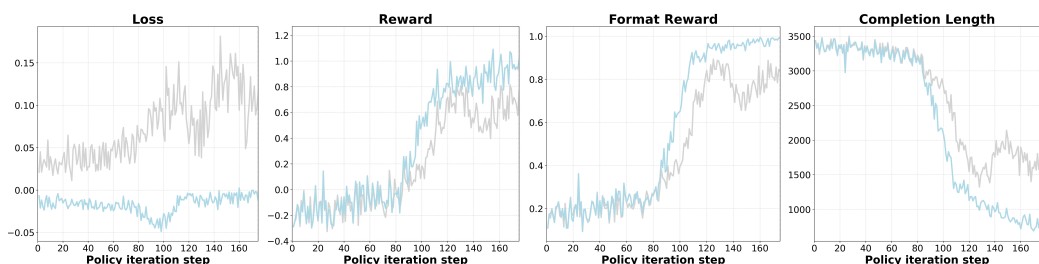

Figure 3: Comparison of GPG(blue curves) and GRPO(gray curves) in terms of training loss, rewards and completion length. Experiments are based on DeepSeek-R1-Distill-Qwen-1.5B, same as Table 3.

## B.6 CASE EXAMPLE

We show a case study in Figure 4.

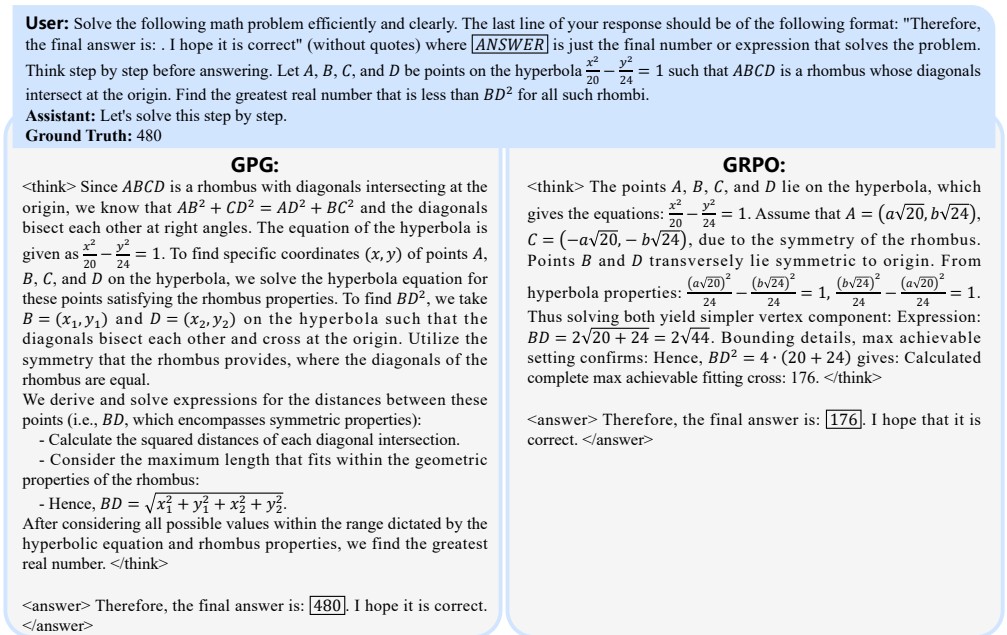

Figure 4: Comparison of GPG and GRPO in mathematical reasoning task based on DeepSeek-R1-Distill-Qwen-1.5B model trained on Open-rs dataset: a test case from AIME24 dataset.

## C MORE RELATED WORK

**Proximal Policy Optimization**. PPO (Schulman et al., 2017) addresses the inherent optimization instability of Trust Region Policy Optimization (TRPO) (Schulman et al., 2015) through a clipped

surrogate objective. Formally, let the probability ratio between the updated policy $\pi_\theta$ and the previous policy $\pi_{\theta_{\text{old}}}$ be defined as

$$r_t(\theta) = \frac{\pi_\theta(a_t|s_t)}{\pi_{\theta_{\text{old}}}(a_t|s_t)}, \tag{18}$$

where $a_t$ and $s_t$ denote the action and state at timestep $t$, respectively. While TRPO maximizes the surrogate objective

$$\mathcal{J}^{\text{TRPO}}(\theta) = \mathbb{E}_t \left[ r_t(\theta) \hat{A}_t \right] \tag{19}$$

under a Kullback-Leibler (KL) divergence constraint, PPO reformulates this via a clipped mechanism. Here, $\hat{A}_t$ represents the estimated advantage function quantifying the relative value of action $a_t$ in state $s_t$. The PPO objective is defined as:

$$\mathcal{J}^{\text{CLIP}}(\theta) = \mathbb{E}_t \left[ \min \left( r_t(\theta) \hat{A}_t, \text{clip}(r_t(\theta), 1 - \epsilon, 1 + \epsilon) \hat{A}_t \right) \right], \tag{20}$$

where the clip operator restricts $r_t(\theta)$ to the interval $[1 - \epsilon, 1 + \epsilon]$, with $\epsilon$ being a hyperparameter controlling the policy update magnitude. This constraint prevents excessive policy deviations that could degrade performance.

To further stabilize training and promote exploration, the composite objective incorporates three components: 1) Clipped policy gradient term $\mathcal{J}^{\text{CLIP}}(\theta)$, 2) Value function loss:

$$\mathcal{L}^{\text{VF}} = \mathbb{E}_t \left[ (V_\theta(s_t) - V_{\text{target}}(s_t))^2 \right], \tag{21}$$

where $V_\theta(s_t)$ is the state-value function estimator and $V_{\text{target}}(s_t)$ denotes the target value computed via temporal-difference methods, 3) Entropy regularization:

$$\mathcal{H}(s_t, \pi_\theta) = - \sum_{a \in \mathcal{A}} \pi_\theta(a|s_t) \log \pi_\theta(a|s_t), \tag{22}$$

with $\mathcal{A}$ being the action space, which prevents premature policy convergence by encouraging stochasticity.

The complete objective integrates these terms as:

$$\mathcal{J}^{\text{PPO}}(\theta) = \mathbb{E}_t \left[ \mathcal{J}^{\text{CLIP}}(\theta) - c_1 \mathcal{L}^{\text{VF}} + c_2 \mathcal{H}(s_t, \pi_\theta) \right], \tag{23}$$

where $c_1 > 0$ and $c_2 > 0$ are coefficients balancing policy optimization, value estimation accuracy, and exploration. Crucially, PPO replaces TRPO's computationally intensive second-order KL constraints with first-order gradient clipping, enabling efficient large-scale implementations while preserving monotonic policy improvement guarantees, as rigorously established through surrogate objective monotonicity analysis (Hsu et al., 2020).

**Group Relative Policy Optimization**. GRPO (Shao et al., 2024) establishes a policy gradient framework that eliminates dependency on explicit value function approximation through comparative advantage estimation within response groups. The method operates by sampling multiple candidate outputs for each input question and constructing advantage signals based on relative rewards within these groups. For a given question $q \sim P(Q)$, the algorithm generates $G$ responses $\{o_1, \ldots, o_G\}$ from the current policy $\pi_{\theta_{\text{old}}}$, then computes token-level advantages using intra-group reward comparisons.

The advantage term $\hat{A}_{i,t}$ for the $t$-th token in the $i$-th response is defined as the deviation from the group average reward:

$$\hat{A}_{i,t} = R(o_i) - \frac{1}{G} \sum_{j=1}^{G} R(o_j), \tag{24}$$

where $R(\cdot)$ denotes the reward model's evaluation. This design inherently aligns with the comparative training paradigm of reward models, which typically learn from pairwise response rankings.

The optimization objective integrates clipped probability ratios with explicit KL regularization. Defining the token-level probability ratio as:

$$r_{i,t}(\theta) = \frac{\pi_\theta(o_{i,t}|q, o_{i,<t})}{\pi_{\theta_{\text{old}}}(o_{i,t}|q, o_{i,<t})}, \tag{25}$$

the clipped surrogate objective constrains policy updates through:

$$\mathcal{J}_{i,t}^{\text{clip}}(\theta) = \min\left(r_{i,t}(\theta)\hat{A}_{i,t},\, \text{clip}(r_{i,t}(\theta), 1 - \epsilon, 1 + \epsilon)\hat{A}_{i,t}\right). \tag{26}$$

Diverging from PPO's implicit KL control via reward shaping, GRPO directly regularizes policy divergence using an unbiased KL estimator:

$$\mathbb{D}_{\text{KL}}\left[\pi_\theta \| \pi_{\text{ref}}\right] = \frac{\pi_{\text{ref}}(o_{i,t}|q, o_{i,<t})}{\pi_\theta(o_{i,t}|q, o_{i,<t})} - \log\frac{\pi_{\text{ref}}(o_{i,t}|q, o_{i,<t})}{\pi_\theta(o_{i,t}|q, o_{i,<t})} - 1, \tag{27}$$

The complete objective combines these components with a regularization coefficient $\beta$:

$$\mathcal{J}^{\text{GRPO}}(\theta) = \mathbb{E}_{q,\{o_i\}}\left[\frac{1}{G|o_i|}\sum_{i,t}\left(\mathcal{J}_{i,t}^{\text{clip}}(\theta) - \beta\mathbb{D}_{\text{KL}}\left[\pi_\theta \| \pi_{\text{ref}}\right]\right)\right]. \tag{28}$$

