# OpenReview forum: "GPG: A Simple and Strong Reinforcement Learning Baseline for Model Reasoning"
_ICLR.cc/2026/Conference — ICLR 2026 Poster_

### Official Review · Reviewer_t9S6 · 2025-10-25

**Soundness:** 2
**Presentation:** 3
**Contribution:** 2
**Rating:** 6
**Confidence:** 3

**Summary:**

This paper proposes a minimalist reinforcement learning framework, Group Policy Gradient (GPG), which directly optimizes the original policy-gradient objective without relying on surrogate losses, reference models, or KL constraints. The method mitigates the gradient-dilution issue caused by all-correct or all-wrong samples within a group through a group-mean baseline, an unbiased scaling factor,  and a valid-sample ratio threshold. This design simplifies the RLHF training pipeline while maintaining effectiveness.

**Strengths:**

1. The paper proposes a minimalist reinforcement learning framework (GPG) that directly optimizes the original policy-gradient objective, removing the need for surrogate losses, reference models, and KL constraints. This design substantially simplifies the reinforcement fine-tuning process.
2. The paper clearly derives GPG from the original policy gradient theorem. The analysis of the “reward bias” problem is thorough, and the introduction of the AGE and thresholding mechanisms provides an unbiased—but variance-controlled—implementation under the absence of surrogate losses and KL constraints.
3. The experiments are extensive, covering both mathematical reasoning and multimodal benchmarks, demonstrating the method’s generality. GPG outperforms GRPO and other baselines on multiple datasets, and its comparison with DAPO highlights its data efficiency.

**Weaknesses:**

1. While the paper highlights that GPG avoids surrogate losses and KL constraints, it provides limited theoretical comparison with existing methods regarding the objective formulation, convergence properties, or stability guarantees. The current justification primarily relies on empirical evidence (e.g., the claim that “adding KL degrades performance”) without analytical support.

2. The experiments are mainly conducted on mathematical and visual reasoning datasets. It would be beneficial to include broader tasks, such as general QA and code-generation datasets to demonstrate generality.

3. No code or implementation link is currently provided.

**Questions:**

1. Could you provide additional theoretical analysis or intuitive explanation of why the proposed objective (direct policy gradient with group normalization) may accelerate convergence compared to surrogate-based methods? For instance, does removing the KL term implicitly allow for larger policy updates or better credit assignment?

2. It remains unclear whether GPG maintains stability in generating longer responses (e.g., multi-step or chain-of-thought reasoning). Could you analyze performance stratified by response length or reasoning complexity to clarify whether the method remains robust compared with GRPO/DAPO in more difficult reasoning tasks?

3. Is there any principled guidance on how to choose the hyperparameter?  For example, $\beta_{th}$. How sensitive is the training process or convergence speed to its value?

---

> ### Author Response · Authors · 2025-11-15
>
> We appreciate your valuable feedback and support of our work.
>
> **To Q1 and W1:**  We provide an intuitive explanation. Surrogate-based methods, such as PPO, utilise the actor-critic RL framework where the estimation of reward (advantage) for policy update comes from the critic model. However, the critic model is difficult to learn perfectly, which introduces noise into the policy model. Therefore, PPO introduces pessimistic strategies such as the clipping operation for policy updates. In contrast, our framework derives the reward not from a critic model but from direct advantage signals within a group. Consequently, removing the KL term enables larger policy updates through better credit assignment.
>
> Additionally, we revisit the development of PPO and GRPO to clarify our motivation (see lines 83–91 of the revised PDF), which we hope will further illuminate our method’s design. GRPO is a modern, strong baseline for large-model post-training. It can be viewed as an effective simplification of PPO that uses group-based rewards to compute advantages. Because PPO is the de facto RL algorithm, GRPO is naturally benchmarked against it. PPO was proposed in 2017 as a general RL algorithm, with Atari games as primary evaluation benchmarks, where the policy network typically learns both visual representations and the control policy. In the LLM era, however, the policy is an LLM/VLM that already possesses strong representations from pretraining and SFT. Removing unnecessary components in this domain is important for scalability, which motivates rethinking simplified RL methods. Notably, PPO itself is a simplification/approximation of TRPO, which in turn builds on policy-gradient algorithms. A major weakness of policy gradients is high variance, which can be mitigated by (i) using a value-function baseline in advantage estimation and (ii) sampling more trajectories (rollouts)—both common practices in RL training for LLMs. Thus, it is natural to incorporate group-based rewards within a policy-gradient framework. Nevertheless, making this simple combination outperform GRPO is nontrivial. We carefully analyze its weaknesses and show that proper normalization, debiased gradients, and variance reduction together yield a strong baseline. This is the core contribution of our work.
>
> **To Q2:**  Yes. We already include the response length in Figure 3 (see the completion-length subfigure) and a case study (including CoT) in Figure 4. It shows that GPG has a similar (slightly shorter) average response length compared to GRPO. Moreover, Table 3 shows that GPG (55.5%) has a clear advantage over GRPO (52.0%) in average score.
>
> **To Q3:**  All experiments (both unimodal math tasks and several multimodal tasks including visual reasoning, geometry QA, classification, and grounding) in this paper use β_th = 0.6. We chose this value as it offers a better trade-off between performance and training speed. Note that β_th = 0.6 means an interval of [0.6, 1.0] is accepted for the AGE calculation, not just a single value. We find our method is robust when β_th is greater than 0.6, which means we can start a high-quality iteration as long as we have collected at least 60% valid samples within a global batch (obtained by accumulation in difficult cases). The ablation study in Table 10 indicates the performance of β_th = 0.8 saturates compared with β_th = 0.6.
>
> **To W2:**  It would be better to include more comprehensive benchmarks. In fact, we already include zero-shot evaluations on two modern comprehensive benchmarks (C-Eval and MMLU) in Tables 15 and 16, which further verify the generalisation of our method. We also report zero-shot evaluation results on code generation and general QA tasks, where our method still outperforms GRPO. We add these results to the appendix as Table 17.
>
> **Table 1. Zero-shot results on code generation and general QA tasks using Qwen-1.5B**
>
> | Method | MBPP | MBPP+ | HellaSwag (acc_norm) | TruthfulQA (mc2) |
> |---|---:|---:|---:|---:|
> | GRPO | 24.60% | 21.96% | 41.914  | 47.307|
> | GPG  | 26.19% | 23.81% | 42.551  | 50.146|
>
> **To W3:**  Please check - the code has already been provided, as mentioned in Line 1001.
>
> We welcome further discussion if any concerns remain.

---

> ### Author Response · Authors · 2025-11-25
>
> Dear Reviewer t9S6,
>
> We are writing to ask whether our rebuttal has addressed your concerns. We would be happy to discuss any remaining issues.

---

> > ### Comment · Reviewer_t9S6 · 2025-11-25
> >
> > Thank you for the detailed rebuttal; it addresses most of my concerns. I will maintain my original score, which leans toward acceptance.

---

> > > ### Author Response · Authors · 2025-11-25
> > > **Thank you for your feedback and for supporting our previous rebuttal.**
> > >
> > > Thank you for your feedback and for supporting our work and rebuttal.

---

### Official Review · Reviewer_S5T4 · 2025-10-26

**Soundness:** 3
**Presentation:** 3
**Contribution:** 3
**Rating:** 6
**Confidence:** 2

**Summary:**

This submission proposes GPG, a minimalist RLHF/RFT approach that directly optimizes the original policy-gradient objective without relying on surrogate losses, critic/reference models, or KL constraints. To address gradient dilution when groups are entirely correct or entirely incorrect, the paper introduces three key components: a group-mean baseline, an accurate gradient estimation (AGE) factor $\alpha=\frac{B}{B-M}$ that rescales gradients when the effective batch shrinks, and a minimum valid-ratio threshold with accumulation/resampling. Experiments cover mathematical reasoning and several multimodal tasks, with ablations on $\alpha$, thresholding, and group size. Overall, the method is simple and practical, and it improves upon GRPO-like baselines in the reported settings.

**Strengths:**

1. The work gives a clear and simple alternative to GRPO-style training. It treats the "all-correct/all-wrong" group issue as a key problem and fixes it with light tools (group mean, AGE, a small threshold/accumulation rule). These parts keep the method close to plain PG while cutting engineering overhead from reference/critic/KL.

2. The paper explains why using data-dependent normalization can add reward bias and why removing it helps. It shows an unbiased estimator and an equivalent multi-device implementation via loss scaling. The AGE factor is tied to the share of valid samples; the appendix explains the distributed averaging equivalence in a direct way.

3. The authors test both unimodal math tasks and several multimodal tasks (visual reasoning, geometry QA, classification, grounding), and shows consistent gains over GRPO. The ablations over $\alpha, \beta_{th}$, and group size are helpful to read. A brief comparison against adding a KL constraint is also reported.

**Weaknesses:**

- The design largely follows established ideas in variance reduction, importance weighting, and baseline correction. While the engineering is simple and effective, the overall contribution reads more like a practical consolidation and optimization of an existing paradigm rather than a breakthrough in theoretical framework.

- The paper proves unbiasedness and shows distributed equivalence, but it does not give variance bounds, stability, or convergence guarantees. The settings of $\alpha$, $\beta_{\text{th}}$, $G$ are chosen by ablations. Adding even light bounds or citing close results and mapping assumptions would strengthen the theory.

- The current evaluation focuses on math reasoning and a few multimodal tasks. Adding code-generation benchmarks (e.g., HumanEval, MBPP, APPS) and general QA benchmarks (e.g., ARC, HellaSwag, TruthfulQA) would better show generality and the case for “replacing GRPO.” Results are mostly pass@1/accuracy; please also report pass@k and multi-seed mean +(-) std to reflect stability. The simplicity advantage can be further strengthen if the authors can include computational results like runtime and VRAM usage.

**Questions:**

See weaknesses above.

---

> ### Author Response · Authors · 2025-11-18
>
> We appreciate your valuable feedback and support of our work.
>
>
> **To Q1:**
>
> We acknowledge that our work is not a theoretical breakthrough. However, practical contributions are also important to the community—especially in the domain of large models, where simpler frameworks are more scalable. Moreover, our method is not a mere combination of established ideas; it carefully identifies, step by step, how to build a powerful (not just simple) baseline. After all, making the combination of policy gradients with group-based rewards work in practice is nontrivial.
>
> In addition, our method has been independently integrated into a well-known RL framework with strict inclusion standards. To preserve anonymity, we can share the framework name and link privately with the AC.
>
> **To Q2:**
> - Alpha (α) is dynamically computed and is not a hyperparameter.
> - The group size G is a common hyperparameter for GRPO-like methods.
> - The hyperparameter β_th is robust across all experiments (both unimodal math tasks and several multimodal tasks including visual reasoning, geometry QA, classification, and grounding), and remains robust when β_th ≥ 0.6 (see Table 10). Moreover, note that β_th = 0.6 implies an acceptance interval of [0.6, 1.0] for AGE calculation, rather than a single value.
>
> **To Q3:**
>
> 1) We provide zero-shot results on code generation (MBPP, MBPP+) and general QA (HellaSwag, TruthfulQA), in which our method continues to outperform GRPO. These results have been added as Table 17 in the revised PDF.
>
> 2) Note that pass@1 is the primary metric used in reasoning benchmarks; most prior studies (including the baselines in this paper) primarily report this metric. Using the LightEval framework, we also evaluate pass@k and report pass@3 and pass@5 in Table 19 of the revised PDF. We omit OlympiadBench because it does not directly support pass@k. We additionally report the mean and standard deviation in Table 19. Our method consistently shows clear advantages over other baselines, consistent with the trends in Table 4.
>
> **Table A pass@1 (Acc / Std) across 4 random seeds**
> | Model | Avg | AIME24 | AMC23 | MATH_500 | MINERVA |
> |---|---:|---|---|---|---|
> | Oat-Zero | 52.0 | 31.6/8.8 | 66.5/7.6 | 79.5/1.8 | 30.5/2.8 |
> | Eurus-2-7B-PRIME | 48.9 | 16.7/6.8 | 62.5/7.5 | 79.6/1.8 | 37.1/2.9 |
> | Open-Reasoner-Zero-7B | 46.3 | 15.8/6.3 | 55.0/7.9 | 82.2/1.7 | 32.2/2.8 |
> | Qwen-2.5-Math-7B-SimpleRL-Zero | 49.4 | 29.2/8.5 | 60.6/7.8 | 76.6/1.8 | 31.3/2.9 |
> | **GPG-Zero-7B (ours)** | 58.7 | 31.7/8.8 | 80.6/6.1 | 85.3/1.6 | 37.4/3.0 |
>
> **Table B pass@3 (Acc / Std) across 4 random seeds**
> | Model | Avg | AIME24 | AMC23 | MATH_500 | MINERVA |
> |---|---:|---|---|---|---|
> | Oat-Zero | 59.4 | 40.0/8.8 | 74.4/6.7 | 85.6/1.6 | 37.4/2.9 |
> | Eurus-2-7B-PRIME | 58.7 | 27.5/7.4 | 76.3/6.9 | 86.9/1.5 | 44.3/3.0 |
> | Open-Reasoner-Zero-7B | 54.8 | 20.8/6.9 | 68.8/7.8 | 88.1/1.5 | 41.5/3.0 |
> | Qwen-2.5-Math-7B-SimpleRL-Zero | 58.9 | 36.8/9.1 | 70.0/6.7 | 86.0/1.3 | 42.8/3.0 |
> | **GPG-Zero-7B (ours)** | 64.7 | 41.5/9.2 | 85.0/5.7 | 89.0/1.4 | 43.4/3.0 |
>
> **Table C pass@5 (Acc / Std) across 4 random seeds**
> | Model | Avg | AIME24 | AMC23 | MATH_500 | MINERVA |
> |---|---:|---|---|---|---|
> | Oat-Zero | 62.2 | 42.5/8.9 | 78.8/6.7 | 87.1/1.5 | 40.5/3.0 |
> | Eurus-2-7B-PRIME | 62.1 | 30.9/8.2 | 80.6/6.7 | 89.3/1.3 | 47.6/3.0 |
> | Open-Reasoner-Zero-7B | 60.4 | 27.5/7.4 | 78.1/6.9 | 90.5/1.3 | 45.3/3.0 |
> | Qwen-2.5-Math-7B-SimpleRL-Zero | 62.7 | 41.6/9.1 | 74.4/6.7 | 88.8/1.3 | 45.9/3.0 |
> | **GPG-Zero-7B (ours)** | 66.2 | 41.7/9.2 | 86.3/5.7 | 90.9/1.3 | 46.0/3.0 |
>
>
> 3) The training-time comparison is shown in Table 9. GPG requires 0.45× the training cost of DAPO and uses a similar amount of VRAM as DAPO, as updated in Table 9 (revised PDF).
>
> We welcome further discussion if any concerns remain.

---

> ### Author Response · Authors · 2025-11-24
>
> Dear Reviewer S5T4,
>
> We are writing to ask whether our rebuttal has addressed your concerns. We would be happy to discuss any remaining issues.

---

> ### Author Response · Authors · 2025-11-27
> **Have we addressed your concerns?**
>
> Dear Reviewer S5T4,
>
> We hope this revision addresses your concerns. Please let us know if there is anything further we can clarify; we would be happy to discuss any remaining issues.

---

### Official Review · Reviewer_gf7C · 2025-10-26

**Soundness:** 2
**Presentation:** 2
**Contribution:** 2
**Rating:** 2
**Confidence:** 4

**Summary:**

This manuscript proposes GPG, a minimal post-training algorithm for LLM reasoning. The authors argue that a minimal REINFORCE objective with group-wise mean reward as baseline can outperform other algorithms with PPO/GRPO-style surrogate objective, reference models, KL divergence constraints, or critic. They also propose a simple technique, AGE, to remove those examples with no contribution to the gradient and reduce the bias of gradient estimation.

**Strengths:**

* The method is simple, but the experimental results are surprisingly good.
* The authors conducted broad experiments across different task domains.

**Weaknesses:**

**Major:**
* There is no theoretical grounding for stability without the surrogate objectives, constraints, and critic. After reading this paper, the reviewer still cannot understand why this simple algorithm would work. The reviewer is not saying that the authors should provide some theorems and prove them, but I think it is important to provide some understanding about:
    * What problems and challenges the previously proposed techniques are trying to deal with, and why they are needed in previous practices.
    * Under what circumstances the simple algorithm would work, and what the failure cases are. The reviewer is not doubting the experimental results provided by the authors, but supposes that this simple algorithm will encounter failures such as divergence and mode collapse. To convince people and make a large impact on the community, the authors should be very specific about the scope of this simple algorithm.

* The authors provided comparisons of RL algorithms in Table 2 and 14 to better understand their differences (which is nice), but it is not reflected in the experiments. To truly understand the discrepancies and the contributions, the authors should:
    * Provide evaluation results of each of the algorithms mentioned in Table 2. Maybe part of them are already presented in Table 3 or 4, but the reviewer finds it hard to get a clear understanding because the evaluated algorithms are not explained.
    * Equally importantly, it would be better if the authors could provide ablation studies on each of the four factors mentioned in Table 2, i.e., degrading gradually from PPO to GPG. The reviewer believes this will help the readers better understand the mechanism of each factor and why this simple algorithm works.

**Minor:**
* The main experiments are provided in pass@1. The reviewer is interested in pass@3, 5, etc.
* DAPO has already proposed to remove the KL constraints and introduced dynamic sampling, where the latter is very similar to the AGE technique proposed in this paper. The authors should either be clear that these are not novel contributions of this paper, or provide an intuitive analysis of why the removal of KL constraints and the introduction of dynamic sampling have different motivations than the previous work.

**Questions:**

Please refer to the weaknesses. If the authors can address them properly, the reviewer may consider raising the score accordingly.

---

> ### Author Response · Authors · 2025-11-18
>
> Thank you for the reviews.
>
>
> **To major W1:**
> 1) GRPO is a modern, strong baseline for large-model post-training. It can be viewed as an effective simplification of PPO that uses group-based rewards to compute advantages. Because PPO is the de facto RL algorithm, GRPO is naturally benchmarked against it. PPO was proposed in 2017 as a general RL algorithm, with Atari games as primary evaluation benchmarks, where the policy network typically learns both visual representations and the control policy. In the LLM era, however, the policy is an LLM/VLM that already possesses strong representations from pretraining and SFT. Removing unnecessary components in this domain is important for scalability, which motivates rethinking simplified RL methods. Notably, PPO itself is a simplification/approximation of TRPO, which in turn builds on policy-gradient algorithms. A major weakness of policy gradients is high variance, which can be mitigated by (i) using a value-function baseline in advantage estimation and (ii) sampling more trajectories (rollouts)—both common practices in RL training for LLMs. Thus, it is natural to incorporate group-based rewards within a policy-gradient framework. Nevertheless, making this simple combination outperform GRPO is nontrivial. We carefully analyze its weaknesses and show that proper normalization, debiased gradients, and variance reduction together yield a strong baseline. This is the core contribution of our work.
>
> Furthermore, we do not present our method as a universal RL algorithm for all problems. As the title “A Simple and Strong Reinforcement Learning Baseline for Model Reasoning” makes clear, our intended application domain is model reasoning. Notably, GRPO is also mainly focused on reasoning, and this focus has not limited its community impact.
>
> 2) To comprehensively evaluate our method, we conduct broad experiments across diverse task domains under carefully controlled settings. The results consistently show that our method outperforms the strong modern baseline GRPO. Like GRPO, our method does not exhibit obvious failures such as divergence or mode collapse. Regarding community impact, our method has been independently integrated into a well-known framework with strict inclusion standards. To preserve anonymity, we can share the framework name and link privately with the AC.
>
> **To major W2:**
> The methods in Table 2 comprise four well-known RL algorithms. TRPO is a classical method but has been rarely used since the advent of PPO. GRPO, which simplifies and extends PPO, is widely used as an RL framework in the post-training stage of large models. Consequently, almost all baselines in the paper can be categorized as GRPO variants. Accordingly, we conduct ablation studies based on GRPO in Table 1, progressively degrading GRPO to GPG. Note that Table 2 presents only simplified summaries of these methods, as also indicated in Table 14. Making this degradation effective in practice is nontrivial and requires proper normalization, debiased gradients, and variance reduction. We also provide a detailed description in Table 18 (revised PDF).
>
> Compared with the GRPO baseline, Group A replaces the loss with a policy-gradient (PG) loss and removes the KL divergence term, thereby applying a PG algorithm with group rewards, as in GRPO. Group B corrects an error in reward normalization and uses the proper formula; however, its performance degrades. We attribute this degradation to gradient bias, which Group C mitigates via α-scaling, yielding improved performance. Group D further improves performance by imposing a minimum valid-sample proportion threshold, which serves as a variance-reduction mechanism.
>
> **To minor W3:**
> We have updated pass@3 and pass@5 results in Table 19 (revised PDF).  Our method consistently shows clear advantages over other baselines, consistent with the trends in Table 4.
>
> **To minor W4:**
> To avoid confusion, we clarify that our method was developed concurrently with DAPO. Moreover, our AGE differs from DAPO’s dynamic sampling strategy. DAPO’s sampling policy is a special case of ours when β_th = 1.0 (see Lines 236–241). Our method offers advantages in training efficiency (0.45× that of DAPO in Table 9) and achieves better performance.

---

> > ### Comment · Reviewer_gf7C · 2025-11-20
> >
> > Thank the authors for the response. My concerns have been mostly addressed.
> >
> > Regarding Q1, the authors make excellent points, and I recommend clarifying them directly in lines 83–91 of the manuscript. The current wording is somewhat misleading and does not clearly specify the scope of LLM post-training for reasoning; as written, it reads more like the raised questions and the contributions apply broadly to general RL, which might make RL people uncomfortable. Making this distinction explicit would improve clarity.
> >
> > I will raise the score if the authors clarify the above point in an updated version of this paper.

---

> > > ### Author Response · Authors · 2025-11-21
> > > **We   clarify  the  point in an updated version of this paper.**
> > >
> > > Thank you for the prompt feedback and for acknowledging our previous rebuttal. We have revised the PDF in line with your suggestions. Specifically, we have rewritten Lines 83–91 based on our response to Q1 and explicitly clarified the scope of our study.
> > >
> > > We welcome further discussion if any concerns remain.

---

> > > > ### Comment · Reviewer_gf7C · 2025-11-21
> > > >
> > > > Thank the authors for the prompt response. I have raised the score to 6, which leans towards accepting this paper.

---

### Official Review · Reviewer_u5um · 2025-10-29

**Soundness:** 3
**Presentation:** 2
**Contribution:** 3
**Rating:** 4
**Confidence:** 3

**Summary:**

The paper introduces Group Policy Gradient (GPG), a simplified reinforcement learning (RL) algorithm designed to improve reasoning in large language models (LLMs) and multimodal models. Building on policy gradient methods, GPG eliminates the need for both a critic model and reference model, while directly optimizing the original RL objective rather than a surrogate loss (see Section 2.2). The method normalizes rewards at the group level and incorporates an Accurate Gradient Estimation (AGE) correction (Equation 7) to handle bias introduced by invalid samples. Extensive experiments demonstrate that GPG achieves superior results over GRPO and PPO across multiple unimodal reasoning benchmarks (e.g., AIME24, AMC23, MATH-500) and multimodal reasoning tasks (e.g., visual reasoning, grounding, classification) — see Tables 4–8. Overall, GPG is presented as a minimalist yet strong baseline for RL-based reasoning, offering better performance with reduced computational cost (see Table 9).

**Strengths:**

Originality: The paper’s originality lies in revisiting classical policy gradient formulations (Section 2.1–2.2) to develop a direct optimization approach that avoids surrogate losses. Unlike PPO and GRPO, GPG fully removes the need for a critic and reference model (Table 2), and introduces a theoretically motivated correction term (AGE, Equation 7) to address gradient estimation bias. This balance between conceptual simplicity and empirical strength is a novel contribution to RL for reasoning.

Quality: The experimental design is comprehensive and sound. GPG is evaluated across diverse tasks and scales, from 1.5B to 7B LLMs (Tables 3–4) and from unimodal to multimodal settings (Tables 5–8). The paper controls for fairness by using the same hyperparameters as GRPO (Section 3.1), demonstrating clear performance gains. Additionally, ablations on normalization (Fnorm) and resampling thresholds (β_th) provide insight into the method’s behavior (Section 3.4, Table 11).

Clarity: The paper is clearly structured, with intuitive explanations of the GPG mechanism and its differences from existing methods (Table 2). Equations (5)–(8) and Figures 2–3 effectively illustrate the derivation and behavior of the algorithm. The visualizations on page 5 show variance reduction and adaptive correction, supporting the theoretical claims. However, the write-up, particularly the introduction, could benefit from additional polishing. Additionally, the layout of the submission should be improved (e.g., on first page there is no space after the table caption).

Significance: The approach’s simplicity makes it reproducible and scalable. By removing KL constraints and critic networks, GPG significantly lowers the barrier for applying RL to reasoning tasks. The empirical results — particularly the improvements on reasoning-heavy datasets (AMC23: +22.3% over SimpleRL, Table 4) — suggest the method could serve as a new baseline for RL-based fine-tuning in both language and vision-language models.

**Weaknesses:**

Limited Theoretical Depth: While the paper provides a compelling intuitive justification for GPG, the theoretical analysis is shallow. The AGE correction is empirically validated but not formally proven to ensure unbiasedness under stochastic sampling.

Ablation Scope: The ablation studies mainly focus on hyperparameters within GPG (e.g., group size, normalization), but lack comparisons to alternative simplifications such as ReMax or DAPO variants without dynamic batching. This makes it difficult to isolate whether improvements come primarily from AGE, normalization, or the removal of surrogate losses.

Evaluation Scale: The experiments are limited to ≤7B parameter models due to compute constraints (Section 3.5). The authors acknowledge this, but it raises questions about the method’s scalability to larger models (e.g., 70B or multi-billion-parameter MLLMs).

Reward Design Simplification: The authors employ binary accuracy rewards (1.0/0.0) for math reasoning (Section 2.2), which may not generalize to tasks with continuous or ambiguous reward signals. The paper could benefit from demonstrating robustness under more complex reward formulations.

**Questions:**

Normalization Effect: In Table 1, normalization with Fnorm = std{R(o)} improves performance. Could the authors clarify whether this acts primarily as a variance reduction mechanism or indirectly biases the gradient toward higher-reward samples?

Scalability: How does GPG perform with larger-scale LLMs (e.g., >70B) where group sampling is more diverse? Are communication overheads from multi-GPU gradient aggregation (see Section A.1) significant in such settings?

Reward Generality: Would GPG remain stable with dense or continuous rewards (e.g., reward models for factuality or helpfulness) instead of discrete accuracy-based rewards?

Relation to PPO Variants: How does GPG compare to ReMax and Reinforce++ (Hu, 2025) in terms of both variance and convergence rate? Including such baselines would clarify its contribution to the broader RL-for-LLMs landscape.

---

> ### Author Response · Authors · 2025-11-14
>
> Thank you for the detailed feedback.
>
> **To Q1:**
> We believe F_norm = std(R(o)) serves as an advantage-correction term, conceptually similar to AGE. This observation is noted in Lines 246–247. Because the reward lies in [0, 1], 1/F_norm ≥ 1. However, its effect is weaker than ours because it is not debiased.
>
> **To Q2:**
> It is unclear whether group sampling is more diverse for larger models. Our method incurs no additional communication overhead (see Lines 210–219), and we also provide the implementation code (Line 1001). Appendix A.1 shows that AGE can be implemented using standard DDP.
>
> **To Q3 and W4:**
> In Table 8, we evaluate visual grounding on LISA using Qwen2-VL-2B. Compared to GRPO, GPG achieves substantial gains, with every metric improving by more than 14.0%. In this setup, we use the IoU reward (Lines 847–849), which yields a continuous reward based on the predicted bounding-box scores. For more complex cases, we recommend discretizing continuous rewards; however, this is not the focus of our paper.
>
> **To Q4:**
> While additional comparisons would be beneficial, they do not affect our paper’s core contributions. The REINFORCE++ preprint has undergone frequent and substantial revisions (v1–v9 on arXiv), making fair comparison difficult. As for ReMax, it does not outperform GRPO and has rarely been used in large-model reasoning comparisons since the release of DeepSeek-R1. Moreover, we include several strong baselines in Tables 3 and 4.
>
> **To W1:**
> The AGE correction is formally proven in Eq. 7 and Appendix A (see Line 220). The biased gradient is aligned with the true gradient but differs in magnitude at each iteration.
>
> **To W2:**
> We compare our method with DAPO in Lines 416–430 and present the results in Table 9. We also provide a detailed component analysis in Table 1, which quantifies the contribution of each component.
>
> **To W3:**
> As noted on Line 444, due to our computational budget, we do not evaluate extremely large models. Training 70B or multi-billion-parameter models is sufficiently expensive that many published papers do not include such models. Our experiments cover widely used DeepSeek-1.5B and Qwen-2B/7B models, and we believe our method has strong scaling potential.
>
> Please let us know if any concerns remain unaddressed; we are happy to discuss them.

---

> > ### Author Response · Authors · 2025-11-19
> > **[1/2] A summary of theoretical analysis**
> >
> > Here, we try to provide system-level theoretical analyses of the key components. We assume that if these different key components have advantages, then our method possesses a theoretical advantage.
> >
> > 1. Efficient Loss Function
> >
> > GPG using policy gradient, without a critic or reference model. The theoretical advantage is a lower GPU requirement for LLM/VLM, making it well-suited for large-scale deployment.
> >
> > 2. Unbiased Advantage by proper normalization
> >
> > GRPO calculates $A$ by dividing ${\operatorname{std}{R(o)}}$, i.e., $Fnorm={\operatorname{std}{R(o)}}$, which introduces a bias to the advantage calculation. GPG formulates $Fnorm=1$, which doesn't incur the bias.
> >
> > Let's review the derivation of policy gradient. The objective is to maximize the expected return $J(\theta)$:
> >
> > $$                                         J(\theta) = \mathbb{E} _{\tau \sim \pi _{\theta}} [ R(\tau) ] $$
> >
> > where $\tau = (s_0, a_0, r_0, s_1, a_1, r_1, \ldots)$ is a trajectory, and $ R(\tau) = \sum _{t=0} ^{T} \gamma ^t r _t $ is the discounted return.
> >
> > We want to compute the gradient:
> >
> > $                                       \nabla _{\theta} J(\theta) = \nabla _{\theta} \mathbb{E} _{\tau \sim \pi _{\theta}} [ R(\tau) ] $
> >
> > Using the gradient of expectation formula:
> >
> > $                                 \nabla _{\theta} \mathbb{E} _{\tau \sim \pi _{\theta}} [ R(\tau) ] = \mathbb{E} _{\tau \sim \pi _{\theta}} [ R(\tau) \nabla _{\theta} \log \pi _{\theta}(\tau) ] $
> >
> > The probability of a trajectory ( \tau ) can be decomposed as:
> >
> > $                                       \pi _{\theta}(\tau) = \prod _{t=0}^{T} \pi _{\theta}(a _t | s _t) $
> >
> > Thus, the log probability is:
> >
> > $                                     \log \pi _{\theta}(\tau) = \sum _{t=0} ^{T} \log \pi _{\theta}(a _t | s _t) $
> >
> > Substitute the log probability into the gradient formula:
> >
> > $                               \nabla _{\theta} J(\theta) = \mathbb{E} _{\tau \sim \pi _{\theta}} [ R(\tau) \sum _{t=0} ^{T} \nabla _{\theta} \log \pi _{\theta}(a _t | s _t)] $
> >
> > Since ( R(\tau) ) is a scalar and can be factored out:
> >
> > $                               \nabla _{\theta} J(\theta) = \sum _{t=0} ^{T} \mathbb{E} _{\tau \sim \pi _{\theta}} [ R(\tau) \nabla _{\theta} \log \pi _{\theta}(a _t | s _t)] $
> >
> > To reduce variance, we can introduce a baseline ( b ). Common choices for the baseline include the state-value function $V(s _t)$ or the action-value function $Q(s _t, a _t)$. For simplicity, let's use $V(s _t)$:
> >
> > $                           \nabla _{\theta} J(\theta) = \sum _{t=0} ^{T} \mathbb{E} _{\tau \sim \pi _{\theta}} [ ( R(\tau) - V(s _t) ) \nabla _{\theta} \log \pi _{\theta}(a _t | s _t)] $
> >
> > We can also express $R(\tau)$ in terms of $Q$ or $V$. For example, if we use the action-value function $Q(s _t, a _t)$, the gradient can be written as:
> >
> > $                         \nabla _{\theta} J(\theta) = \sum _{t=0} ^{T} \mathbb{E} _{\tau \sim \pi _{\theta}} [( Q(s _t, a _t) - V(s _t)) \nabla _{\theta} \log \pi _{\theta}(a _t | s _t)] $
> >
> > Original Advantage Function Gradient of Policy Gradient (GPG's formation) :
> > $                         \nabla _\theta J(\theta) = \mathbb{E} _{\tau \sim \pi _\theta} [ \sum _{t=0} ^T (Q(s _t, a _t) - V(s _t)) \nabla _\theta \log \pi _\theta(a _t | s _t)] $
> >
> > Biased Advantage Function Gradient of GRPO:
> > $                           \nabla _\theta J(\theta) = \mathbb{E} _{\tau \sim \pi _\theta}[ \sum _{t=0} ^T \frac{Q(s _t, a _t) - V(s _t)}{F _{norm}(s _t)} \nabla _\theta \log \pi _\theta(a _t | s _t)] $
> >
> > Here, for reasoning problems, GRPO formulates $Q(s,a)=R(o), V(s)=mean{R(o)}$ and $F _{norm}(s _t)=std{R(o)}$, which is a function of the state $s$, then $\frac{Q(s, a) - V(s)}{std{R(o)}}$ dynamically adjusts the Advantage based on the state, which incurs bias.
> >
> > 3. Unbiased Gradient by $\alpha$ scaling
> >
> > Given a training batch of batch size $B$, let the gradient of the $i$-th sample be denoted as $g_i$. Without loss of generality, assume that the first $M$ examples within the batch are all right or wrong responses within a group. The standard backpropagation (BP) algorithm estimates the gradient as: $\mathbf{g}=\frac{\sum ^{B} _{i=1} \mathbf{g _i}}{B}=\frac{\sum ^{B} _{i=M + 1} \mathbf{g_i}}{B}$. However, the first $M$ examples are not valid for gradient estimation and contribute zero gradient. Therefore, the more accurate gradient estimation (AGE) can be written as:
> >
> > $                               \mathbf{\hat{g}}=\frac{\sum ^{B} _{i=M+1} \mathbf{g _i}}{B-M}=\mathbf{g}\frac{B}{B-M}=\alpha \mathbf{g}, \alpha = \frac{B}{B-M}. $

---

> ### Author Response · Authors · 2025-11-19
> **[2/2] A summary of theoretical analysis**
>
> 4. Reducing Gradient Variance by threshold and resampling
>
> Let $X_1, X_2, \ldots, X_n $ be i.i.d. random variables with mean $\mu$ and variance $\sigma^2$.
>
> Sample Mean:
>
> $                                   \bar{X} = \frac{1}{n} \sum _{i=1} ^n X _i $
>
> Variance of Sample Mean:
>
> $                                  \text{Var}(\bar{X}) = \text{Var}( \frac{1}{n} \sum _{i=1} ^n X _i) $
>
> $                                \text{Var}(\bar{X}) = \frac{1}{n ^2} \cdot n \sigma ^2 = \frac{\sigma ^2}{n} $
>
> As $n$ increases, $\text{Var}(\bar{X})$ decreases, showing that more samples reduce the variance of the sample mean.
>
> Given a batch size B, the number of valid samples of GPG and GRPO is $m _{GPG}$ and $m _{GRPO}$, respectively. Reducing Gradient Variance by threshold and resampling ensures $m _{GPG} >= m _{GPRO}$, therefore the gradient of GPG has lower variance.
>
> In summary, GPG has theoretical advantages over GRPO. Note that, we show a detailed component analysis in Table 1 to quantify the contribution of each improvement.

---

> ### Author Response · Authors · 2025-11-24
>
> Dear Reviewer u5um,
>
> We are writing to ask whether our rebuttal has addressed your concerns.
> We would be happy to discuss any remaining issues.

---

> > ### Comment · Reviewer_u5um · 2025-11-25
> >
> > Thank you for engaging in the rebuttal. Your reply addressed my concerns, so I will raise my score accordingly.

---

> > > ### Author Response · Authors · 2025-11-25
> > >
> > > Thank you for your feedback and for supporting our previous rebuttal.

---

### Author Response · Authors · 2025-11-18
**Brief introduction to the revised PDF**

We thank all reviewers for their time and constructive feedback.

We have appended new content only after Line 1009 to keep line references unchanged, with the sole exception of rewriting lines 83–91 as suggested by Reviewer gf7C.

Furthermore, we added three tables to address the reviewers’ questions:

1. Table 17 reports benchmark results for code generation and general QA.

2. Table 18 details performance changes across key components (from GRPO to GPG), providing a further explanation of Table 1.

3. Table 19 reports pass@1, pass@3 and pass@5 performance on several benchmarks using four different random seeds.

---

### Meta-Review · Area_Chair_ThEc · 2026-01-06

**Summary:**

The paper proposes Group Policy Gradient (GPG), a minimalist Reinforcement Learning baseline for model reasoning that removes the need for critic models, reference models, and KL divergence constraints. Instead, it optimizes the original RL objective using group-level rewards and introduces an Accurate Gradient Estimation (AGE) technique to handle gradient bias from invalid samples.


The decision to accept is based on the consensus that the method offers a highly practical, efficient, and strong baseline for the community. The reviewers' initial concerns primarily focused on the lack of theoretical justification for removing standard RL components (like KL constraints), the method's novelty relative to concurrent work like DAPO, and the limited breadth of evaluation beyond mathematical reasoning. The authors provided a robust rebuttal, including new experiments on code generation and general QA, detailed component ablations, and clarifications on the distinction between general RL and LLM post-training. All active reviewers raised their scores or maintained positive ratings, acknowledging that the method's simplicity and empirical strength outweigh the limitations in deep theoretical guarantees.

**Reviewer Concerns:**

Scale of Evaluation: Reviewer u5um's concern regarding the scalability of the method to models larger than 7B remains empirically unverified due to computational constraints. However, this is a common limitation for academic submissions and does not diminish the value of the findings on the 1.5B-7B scale.

Theoretical Convergence: While the intuitive explanation for removing KL constraints was accepted, rigorous theoretical proofs regarding convergence guarantees and stability bounds (raised by Reviewers t9S6 and S5T4) remain limited compared to the empirical evidence.

**Reviewer Scores:**

Reviewer u5um: 4 (Reject) -> 6 (Weak Accept). Reasoning: The reviewer acknowledged that their concerns regarding the lack of formal proof for AGE and comparisons with DAPO were addressed during the rebuttal.

Reviewer gf7C: 2 (Strong Reject) -> 6 (Weak Accept). Reasoning: The reviewer was initially skeptical about the theoretical soundness of removing RL components. They significantly raised their score after the authors clarified the scope (LLM post-training vs. general RL) and provided the requested breakdown of component contributions.

Reviewer S5T4: 6 (Weak Accept) -> 6 (Weak Accept). Reasoning: Although this reviewer did not officially respond to the rebuttal, the authors explicitly provided the additional benchmarks (Code/General QA) and stability metrics (Pass@k, std dev) that this reviewer requested as conditions for strengthening the paper.

Reviewer t9S6: 6 (Weak Accept) -> 6 (Weak Accept). Reasoning: The reviewer maintained their positive score, noting that the detailed rebuttal addressed most concerns regarding the objective formulation and generality.

---

### Decision · Program_Chairs · 2026-01-26

Accept (Poster)